# Portfolio Optimization with Sector Return Prediction Models

**Wolfgang Bessler [1,*] and Dominik Wolff [2,3,4]**

1    Faculty of Business Administration, University of Hamburg, Hamburg 20148, Germany
2    Department of Business and Law, Frankfurt University of Applied Sciences,
     60318 Frankfurt am Main, Germany
3    Department of Business Administration, Economics and Law, Technical University Darmstadt,
     64289 Darmstadt, Germany
4    Deka Investment GmbH, 60439 Frankfurt, Germany
*    Correspondence: wolfgang.bessler@uni-hamburg.de

**Abstract:** We analyze return predictability for U.S. sectors based on fundamental, macroeconomic, and technical indicators and analyze whether return predictions improve tactical asset allocation decisions. We study the out-of-sample predictive power of individual variables for forecasting sector returns and analyze multivariate predictive regression models, including OLS, regularized regressions, principal component regressions, the three-pass regression filter, and forecast combinations. Using an out-of-sample Black–Litterman portfolio optimization framework and employing predicted returns as investors' 'views', we evaluate the benefits of sector return forecasts for investors. We find that portfolio optimization with sector return prediction models significantly outperforms portfolios using historical averages as well as passive benchmark portfolios.

**Keywords:** portfolio optimization; Black–Litterman model; return forecasts; predictive regression; three-pass regression filter

**JEL Classification**: G17; G11; C53

## 1. Introduction

Most of the earlier literature on stock return predictability and mutual fund performance concludes that stock returns are hardly predictable and that active asset managers seldom achieve persistently higher risk-adjusted returns. Besides the earlier efficient market explanation for this phenomenon, one of the more recent rationales is the well-functioning equilibration processes in financial markets (Berk and Green 2004; Bessler et al. 2018). Consequently, on average, investors should not expect a superior performance after fees from implementing active portfolio management approaches. The recent popularity of passive investment vehicles such as index funds and ETFs is in accordance with this academic evidence.

However, some findings in the literature provide evidence in favor of active management. First, several studies indicate some out-of-sample predictability for the overall U.S. stock market, offering economic value for investors (e.g., Rapach et al. 2010; Neely et al. 2014). In addition, Bessler et al. (2017) find that employing sophisticated asset allocation models in an out-of-sample setting can generate superior risk-adjusted returns relative to passive investments. In this study, we combine these two ideas first by predicting sector returns and second by using these return predictions for out-of-sample Black–Litterman portfolio optimization. We evaluate the risk-adjusted returns of this strategy by comparing it to passive investments and to allocation strategies based on naïve predictions. Based on the premise that markets are efficient, none of the prediction and optimization models should provide significant return predictability or accomplish higher risk-adjusted returns compared to a passive benchmark portfolio.

Our study contributes to the literature in four major aspects. First, we analyze return predictability for different sectors, while earlier studies mainly focus on the overall U.S. stock market (S&P500). Our premise is that different risk factors determine sector returns, and consequently, the latter diverge substantially during the economic cycle. From a portfolio management perspective, performance should improve when investing and diversifying into different sectors and by adjusting the asset allocation between sectors over time (Grinold and Kahn 2000).

Second, we expand the commonly used dataset of predictive variables (Goyal and Welch 2008) by including additional macroeconomic and technical indicators. Third, we analyze the predictive power of univariate and multivariate prediction models, including established approaches such as principal components, forecast combination models, and selection via the LASSO. We add to the literature by also testing a relatively new target-relevant latent factor approach and proposing a variable selection model. Finally, we analyze the performance of an investment strategy based on return predictions.

Overall, our empirical results suggest that for most sectors, a return forecast model predicts future returns significantly better than the historical average. Moreover, we find that asset allocations based on predicted returns not only significantly outperform the value-weighted market index and the equally weighted (1/N) buy-and-hold portfolio but also outperform asset allocations based on historical averages.

However, our results do not imply that asset managers can easily achieve an outperformance. The main detriments are equilibrium mechanisms in capital markets (Berk and Green 2004) such as fund flows and manager changes (Bessler et al. 2018) that all drive the initially successful strategies to an average performance. Whether implementing our approach is able to generate superior performances in the future is an interesting issue, as the ideas are now publicly available. At least some recent research suggests that there is a post-publication decline in long/short anomaly returns in the United States (Jacobs and Müller 2020).

We organize the remainder of this study as follows. In Section 2, we discuss the literature on portfolio optimization and on stock return predictability. The data are presented in Section 3. We separate our empirical analysis into two major parts. In the first part (Section 4), we investigate the out-of-sample predictive power of macroeconomic, fundamental, and technical indicators as well as the predictability of sector returns with different state-of-the-art multivariate forecast models. In the second part (Sections 5 and 6), we analyze the out-of-sample performance of various optimization models, including the Black–Litterman model, in which we include the sector return forecasts. Section 6 provides an extensive robustness analysis. Section 7 concludes the study.

## 2. Literature Review

In this study, we combine two different strands of the literature: portfolio optimization and return prediction models. We first focus on portfolio optimization and review the related literature in Section 2.1. Subsequently, we discuss the literature on return predictability and the associated predictors in Section 2.2.

### 2.1. Portfolio Optimization Models

For investors, it is crucial to ascertain whether return forecast models result in higher risk-adjusted returns when utilised in asset allocation decisions. Most studies analyzing the economic value of equity market forecasts are based on the Markowitz (1952) mean-variance (MV) framework to compute the optimal portfolio weights for a two-asset portfolio consisting of the U.S. stock market (S&P500) and the risk-free rate. However, the traditional Markowitz (1952) optimization framework usually performs poorly in portfolios with more than two assets due to estimation error maximization (Michaud 1989), corner solutions (Broadie 1993), and extreme portfolio reallocations (Best and Grauer 1991). The literature proposes several variations and extensions of MV, which range from imposing portfolio constraints (Frost and Savarino 1988; Jagannathan and Ma

2003; Behr et al. 2013) to Bayesian methods for estimating the MV input parameters (Jorion 1985, 1986; Pastor 2000; Pastor and Stambaugh 2000). DeMiguel et al. (2009) find that for historical average return forecasts, no optimization model outperforms a naïve diversified 1/N benchmark. In contrast, Bessler et al. (2017) provide evidence that the Black–Litterman (BL) model significantly outperforms MV and 1/N for multi-asset portfolios because it accounts for uncertainty in the return forecast. Consequently, we primarily rely on the BL model to compute optimal asset allocations based on return forecasts. As a robustness check, we also compute traditional Markowitz (1952) and Bayes-Stein (Jorion 1985, 1986) portfolios and compare them to the BL model.

While Markowitz mean-variance optimization implicitly assumes perfect forecasts with no estimation error, the BL model accounts for estimation error in return forecasts and allows for the inclusion of the reliability of each forecast, which we measure as the historical mean squared forecast error (MSFE). The BL model employs a reference portfolio (benchmark) in which the model invests if forecast errors for all assets are (infinitely) large. With increasing forecast precision, the BL optimal portfolio weights deviate more strongly from the benchmark. For perfect forecasts, the BL optimal weights converge to the Markowitz portfolio weights. Moreover, the BL model combines return forecasts ('views') with 'implied' returns. Implied returns are computed from the portfolio weights of the reference portfolio using a reverse optimization technique (Black and Litterman 1992). The combined return estimate is a matrix-weighted average of 'implied' returns and forecasts incorporating the reliability of each forecast. It is computed as follows:

$$\hat{\mu}_{BL} = [(\tau\Sigma)^{-1} + P'\Omega^{-1}P]^{-1}[(\tau\Sigma)^{-1}\Pi + P'\Omega^{-1}Q], \tag{1}$$

in which $\Pi$ is the vector of implied asset returns, $\Sigma$ is the covariance matrix, and Q is the vector of the return estimates. $\Omega$ is a diagonal matrix and contains the reliability of each forecast. P is a binary matrix, which contains the information regarding for which asset a return forecast is employed.[1] The parameter $\tau$ controls the level of deviation from the reference portfolio. The posterior covariance matrix is derived as follows (Satchell and Scowcroft 2000):

$$\Sigma_{BL} = \Sigma + [(\tau\Sigma)^{-1} + P'\Omega^{-1}P]^{-1}. \tag{2}$$

After computing combined return estimates and the posterior covariance matrix, we conduct a traditional risk–return optimization, maximizing the investor's utility, as presented in in the following equation:

$$\max_{\omega} U = \omega'\mu_{BL} - \frac{\delta}{2}\omega'\Sigma\omega, \tag{3}$$

where $\mu_{BL}$ is the BL combined return forecast, $\omega$ is the vector of portfolio weights, $\delta$ is the risk-aversion coefficient, and $\Sigma$ is the covariance matrix of asset returns. There have been other recent advances in portfolio optimization models. Similar to Blackand Litterman (1992), Pedersen et al. (2021) proposes 'enhanced portfolio optimization' to make portfolio optimization work in practice by accounting for noise in the investor's estimates of risk and expected return. James et al. (2023) propose a semi-metric portfolio optimization method with the objective of reducing simultaneous asset shocks in the portfolio. Butler and Kwon (2023) present a stochastic portfolio optimization framework for integrating regression prediction models in a mean-variance optimization (MVO) setting. Testing these models for our prediction framework is left for future research.

### 2.2. Stock Return Predictability

Stock return predictability is a controversially debated issue in the asset management literature. Several studies identify fundamental and macroeconomic variables as well as technical indicators that provide predictive power in forecasting the U.S. equity risk

premium.[2] Among the most prominent predictive variables are the dividend yield, the book-to-market ratio, the term spread, the default spread, the price-to-earnings ratio, the inflation rate, and the stock variance.[3] While earlier studies mainly build on in-sample predictive regressions to identify forecast ability, Goyal and Welch (2008) revisit the predictive power of 14 fundamental and interest-rate-related variables for forecasting the U.S. equity premium out of sample for the period 1927 to 2005. Obviously, out-of-sample predictability is more relevant for investors than in-sample analyses because significant in-sample predictability does not imply that return forecasts are useful to generate a superior portfolio performance[4]. Nevertheless, Goyal and Welch (2008) suggest that none of the fundamental variables proposed in the literature have superior out-of-sample forecast capabilities compared to the simple historical average return in univariate regressions and in several multivariate models. Their results suggest that the market is information efficient and stock returns are unpredictable.

However, several subsequent studies using the same dataset implement more elaborate multivariate return forecast models and report superior out-of-sample forecasts compared to the historical average, thus providing economic benefits to investors (Rapach et al. 2010; Cenesizoglu and Timmermann 2012). Among these models are forecast combination models (Rapach et al. 2010), which combine forecasts of univariate regressions,[5] economically motivated model restrictions[6] (Campbell and Thompson 2008; Pettenuzzo et al. 2014), and predictive regressions based on principal components (Ludvigson and Ng 2007; Neely et al. 2014; Hammerschmidt and Lohre 2018).[7] In addition, Neely et al. (2014) suggest that adding technical indicators to the fundamental predictive regression models improves equity return forecasts for the U.S. stock market. Hammerschmidt and Lohre (2018) report improved predictability for the same dataset by including macroeconomic regime indicators, reflecting the current state of the economy (regime). When judging these results, it is essential to recognize that even a low level of return predictability enables investors to improve their asset allocation decisions (Campbell and Thompson 2008).

However, the vast majority of studies analyze return predictions only for the overall U.S. stock market (S&P500) and investigate performance gains only for a two-asset portfolio consisting of the U.S. stock market (S&P500) and the risk-free rate (Goyal and Welch 2008; Rapach et al. 2010; Cenesizoglu and Timmermann 2012; Neely et al. 2014; Hammerschmidt and Lohre 2018). In our study, we focus on sector return forecasts, as we expect sector returns to deviate substantially from each other during the economic cycle, offering benefits from shifting funds between different sectors over time based on current market conditions. So far, only very few studies forecast sector returns. Ferson and Harvey (1991) and Ferson and Korajczyk (1995) analyze in-sample predictability of sector returns for a small set of lagged predictive variables. We expect to provide superior sector-level forecasts and consequently enhanced portfolio benefits for investors when using fundamental and macroeconomic variables as well as technical indicators simultaneously.

A newer strand of the literature uses machine learning models for stock market forecasting. Rapach et al. (2019) use a machine learning approach to predict industry returns based on the information contained in lagged industry returns from across the entire economy, finding that out-of-sample industry return forecasts are economically valuable. Gu et al. (2020) employ machine learning for a large set of stock-level data to predict individual stock returns, which they subsequently aggregate to index predictions with promising results. Wolff and Neugebauer (2019) analyze predictions of monthly S&P 500 returns with tree-based machine learning models and regularized regression models. At a much higher data frequency, Li et al. (2023) predict ten-minute stock prices of the constituents of the Taiwan 50 Index based on spatiotemporal deep learning.

### 3. Data

In this section, we present the sector data (Section 3.1.) and describe the predictive variables (Section 3.2.) that we employ to compute sector return forecasts.

### *3.1. Sector Data*

We use the following six different sector indices based on data from Thomson Reuters Datastream that begins in 1973 Department of Business Administration, Economics and Law:[8] 'Oil and Gas', 'Manufacturing', 'Consumer Goods and Services', 'Health Care', 'Technology and Telecommunication', and 'Financials'. To compute technical indicators, one year of data is required. Therefore, our evaluation period—for the in-sample analysis—ranges from January 1974 to December 2013 (480 monthly observations). Table 1 Panel A presents summary statistics of monthly sector returns for the full sample. Health Care displays the highest average monthly returns (0.99%), followed by Oil & Gas (0.98%), while Consumer Goods & Services and Financials provide the lowest average returns with 0.88% and 0.90%, respectively. All sector return time series exhibit a negative skewness and a substantial level of excess kurtosis so that we can reject all null hypotheses of normally distributed stock returns. The correlation matrix in Table 1 Panel B indicates significantly positive correlations among all sector index returns with inter-sector correlation coefficients ranging from 0.44 to 0.85, thus, offering only moderate diversification opportunities. Table 1 Panel C provides the market value weights of the sector indices in percent of the overall market as of 31 December 2013.

**Table 1.** Descriptive statistics for sector indices.

| | Market | Oil and Gas | Manufact | Con Gds and Sv | Health Care | Tech & Tele | Financials |
|---|---|---|---|---|---|---|---|
| **Panel A: Summary Statistics** | | | | | | | |
| Mean | 0.89% | 0.98% | 0.96% | 0.88% | 0.99% | 0.91% | 0.90% |
| Median | 1.24% | 1.09% | 1.05% | 1.10% | 1.23% | 1.20% | 1.33% |
| Max | 16.17% | 19.42% | 15.08% | 18.38% | 24.37% | 21.37% | 24.11% |
| Min | −23.26% | −21.02% | −21.52% | −31.67% | −19.57% | −26.98% | −24.06% |
| Std. dev. | 4.53% | 5.54% | 4.66% | 5.16% | 4.59% | 5.83% | 5.77% |
| Skewness | −0.711 | −0.217 | −0.558 | −0.736 | −0.283 | −0.627 | −0.600 |
| Kurtosis | 5.585 | 4.259 | 5.704 | 6.737 | 5.730 | 5.347 | 5.730 |
| Jarque–Bera | 174.42 *** | 35.55 *** | 171.46 *** | 323.36 *** | 155.85 *** | 141.83 *** | 178.3 *** |
| Obs | 481 | 481 | 481 | 481 | 481 | 481 | 481 |
| **Panel B: Correlation Matrix** | | | | | | | |
| Market | 1 | | | | | | |
| Oil and Gas | 0.66 *** | 1 | | | | | |
| Manufact | 0.93 *** | 0.67 *** | 1 | | | | |
| Con Gds and Sv | 0.91 *** | 0.49 *** | 0.85 *** | 1 | | | |
| Health Care | 0.80 *** | 0.44 *** | 0.71 *** | 0.72 *** | 1 | | |
| Tech and Tele | 0.86 *** | 0.46 *** | 0.74 *** | 0.76 *** | 0.56 *** | 1 | |
| Financials | 0.86 *** | 0.51 *** | 0.83 *** | 0.80 *** | 0.74 *** | 0.63 *** | 1 |
| **Panel C: Relative Market Value of Sector Indices Compared to Market Value of Market Index** | | | | | | | |
| Index weight | 100.00% | 10.81% | 18.22% | 24.69% | 11.17% | 17.11% | 18.00% |

Notes. Panel A provides summary statistics of the sector index returns during the period from January 1974 to December 2013 covering 480 months. Panel B provides the correlation matrix for the sector index returns during the period from January 1974 to December 2013. *** indicates values significantly different from 0 at the 10%, 5%, and 1% level, respectively. Panel C provides the relative market value of the sector indices compared to the market value of the overall market index as of 31 December 2013.

*3.2. Predictive Variables*

For forecasting sector and overall stock market returns, we include 19 predictive variables. Due to data availability, our study spans the period from 1973 to December 2013. Table 2 contains a description of the variables together with their abbreviations and data sources. The predictive variables are grouped into fundamental and interest-rate-related variables (Panel A), macroeconomic variables (Panel B), and technical indicators (Panel C).

**Table 2.** Predictive variables.

| Panel A: Fundamental and Interest-Related Variables | | | |
|---|---|---|---|
| **Variable** | **Abb.** | **Source** | **Description** |
| Dividend yield | DY | Thomson Reuters Datastream | Log change in the sector dividend yield. The sector dividend yield is the market-value-weighted average of companies' dividend yields in the sector index. |
| Earnings-to-price ratio | EP | Thomson Reuters Datastream | Log change in the reciprocal of the sector price-to-earnings ratio (PE) The sector PE ratio is the market-weighted average of companies' price-to-earnings ratios in the sector index. |
| Stock variance | SVAR | Own calculation based on Datastream data | Sum of squared daily returns of the sector indices. Daily sector price data are from Thomson Reuters Datastream. |
| Long-term return | LTR | Goyal and Welch (2008) ext. dataset | Return of long-term U.S. government bonds. |
| Term spread | TMS | Goyal and Welch (2008) ext. dataset | Difference between the long-term U.S. government bond yield and the U.S. T-bill rate. |
| Default return spread | DFR | Goyal and Welch (2008) ext. dataset | Return of long-term AAA- and BAA-rated corporate bonds minus the long-term government bond return. |
| Panel B: Macroeconomic Variables | | | |
| Variable | Abb. | Source | Description |
| Inflation | INFL | St. Louis Fed's FRED database | Continuous growth rate of the consumer price index (CPI) for urban consumers. |
| Unemployment claims | CLAIMS | St. Louis Fed's FRED database | Log change in the continuous growth rate of the seasonally adjusted number of initial claims of unemployment. |
| Industrial production | INDP | St. Louis Fed's FRED database | Continuous growth rate of the U.S. industrial production index. |
| Chicago Fed National Activity Index | CFNAI | St. Louis Fed's FRED database | Monthly indicator of overall economic activity. It is compiled by the Chicago Fed based on 85 existing monthly indicators of economic activity. It has a target average value of zero and a target standard deviation of 1. A positive (negative) CFNAI value indicates growth above (below) the trend growth in the economy. |
| Building permits | PERMITS | St. Louis Fed's FRED database | Continuous growth rate of new private housing units in the U.S. that have received building permits (seasonally adjusted). |
| Trade-weighted U.S. dollar index | FEX | St. Louis Fed's FRED database | Continuous growth rate of the dollar index, which is a trade-weighted average of the foreign exchange value of the U.S. dollar against a subset of the broad index currencies including the Euro Area, Canada, Japan, United Kingdom, Switzerland, Australia, and Sweden. |

| Oil price | OIL | Thomson Reuters Datastream | Continuous growth rate of the crude oil price (BRENT). |
|---|---|---|---|
| Panel C: Technical Indicators | | | |
| Variable | Abb. | Source | Description |
| Moving average | MA | Own calculation based on Datastream data | Indicator that compares the actual sector index prices with a 12-month moving average of the previous sector index prices. It generates a buy (1) signal if the actual sector index is above or equal to the long-term moving average and a sell (0) signal if the actual sector index is below the long-term moving average. |
| Momentum | MOM | Own calculation based on Datastream data | Indicator generating a buy signal (1) if the actual sector index trades above or is equal to the level the index traded at 12 months ago and a sell (0) signal if the index currently trades below the level it traded at 12 months ago |
| Volume-based signal | VOL | Own calculation based on Datastream data | Signal that builds on the recent trading volume of one sector index and the direction of price change of the index. More specifically, it uses the 'on-balance' volume (OBV), which is the product of trading volume during a month and the direction of price change during the same month. The volume-based signal generates a buy signal (1) if the actual OBV is above or equal to the 12-month moving average of the OBV and a sell signal (0) otherwise. |
| Relative strength | RS | Own calculation based on Datastream data | Is computed as the return of a sector index during the last six months minus the return of the market index during the last six months. |
| Relative strength index | RSI | Own calculation based on Datastream data | Builds on daily price data of an index during the last month. It is the sum of price changes on days with positive returns divided by the total sum of absolute price changes. |
| Lagged stock market return | MARKET | Own calculation based on Datastream data | Continuous stock market return in the last month. |

The group of fundamental and interest-rate-related variables is widely tested in the literature (Goyal and Welch 2008).[9] Based on previous empirical evidence, we employ the sector dividend-yield and the sector earnings-to-price ratios as fundamental variables, reflecting the sector profitability and providing some predictive power for the overall stock market (Dow 1920; Fama and French 1988, 1989; Ang and Bekaert 2007). We include the variance of daily sector returns, as volatility has some predictive power for the U.S. stock market (Guo 2006). The employed interest-rate-related variables are the returns of long-term U.S. government bonds, the term spread, and the default spread. The term structure of interest rates contains implied forecasts of future interest rates, and the term spread effectively predicts stock returns (Campbell 1987). The default spread also offers predictive power (Keim and Stambaugh 1986; Fama and French 1989) because it usually widens during economic recessions and narrows during expansions due to changes in (perceived) default risk.

The group of macroeconomic variables includes the inflation rate, unemployment claims, industrial production, the Chicago Fed National Activity Index, building permits, the trade-weighted dollar index, and the oil price. All macroeconomic variables are indicators for the overall state of the economy.[10] There is evidence that common stock returns and inflation are negatively correlated (Nelson 1976; Fama 1981). Unemployment

claims are an early indicator for the job market. They usually increase during economic recessions and therefore negatively correlate with future stock returns. The industrial production index measures real output for all facilities located in the United States, including manufacturing, mining, electric, and gas utilities. It is an indicator for industrial growth and therefore positively related to sector stock returns. The Chicago Fed National Activity Index (CFNAI) is designed to gauge overall economic activity by weighting 85 monthly national economic activity indicators. A positive (negative) index indicates growth above (below) the trend.[11] Because the housing market is generally the first economic sector to rise or fall when economic conditions improve or deteriorate, building permits are supposed to be a valuable early indicator for the overall stock market and sector returns. The trade-weighted dollar index reflects the strength of the dollar relative to major foreign currencies, with changes affecting the export and import activity of U.S. companies and subsequently revenues and stock prices. Oil is an important production and cost factor, and a declining oil price usually increases company earnings, economic growth, and stock prices (Driesprong et al. 2008).

Inflation, the industrial production index, and building permits are available on a monthly basis for the previous month. We include two lags to avoid any forward-looking bias. The Chicago Fed publishes the National Activity Index for the previous or antepenultimate month. We include three lags for the CFNAI to make sure that the computation of the forecasts includes data available at each point in time. The St. Louis Fed's FRED publishes the initial claims of unemployment and the trade-weighted U.S. dollar index on a weekly basis. We lag these by one month in line with the fundamental and technical variables.

The third group of predictive variables includes technical indicators using information on investor behavior. Neely et al. (2014) suggest that adding technical indicators improves return forecasts for the overall U.S. stock market (S&P 500). As technical trend-following indicators, we employ moving averages, momentum, and volume-based signals for forecasting sector returns (Neely et al. 2014; Sullivan et al. 1999), as well as the relative strength and the relative strength index (Wilder 1978).

Table 3 Panel A provides summary statistics for the monthly predictive variables for the period from December 1973 to December 2013. Note that fundamental and technical variables are distinct for each sector. For brevity, we only present summary statistics for the fundamental and technical variables for the Oil and Gas sector, as these are quite similar for other sectors. We use the log difference for virtually all fundamental and macroeconomic variables. Table 3 Panel B presents the correlation matrix of predictive variables. All correlation coefficients have an absolute value below 0.85, indicating that the predictive variables capture different information.

**Table 3.** Descriptive statistics for predictive variables.

| Panel A: Summary Statistics | | | | | | | | | | | | | | | | | |
|---|---|---|---|---|---|---|---|---|---|---|---|---|---|---|---|---|---|
| Fundamental and Interest-Rate Related | | | | | | | Macroeconomic Variables | | | | | | Technical Indicators | | | | |
| | DY | EP | SVAR | LTR | TMS | DFR | INFL | CLAIMS | INDP | CFNAI | PERMIT | FEX | OIL | MA | MOM | VOL | RSI | RS | MKT |
| Mean | −0.10% | −0.10% | 0.40% | 0.70% | 2.10% | 0.00% | 0.30% | 0.00% | 0.20% | −8.00% | −0.10% | −0.10% | 0.70% | 74.60% | 77.50% | 58.00% | 3.40% | 0.10% | 0.90% |
| Median | −0.40% | −0.30% | 0.30% | 0.80% | 2.30% | 0.10% | 0.30% | −0.30% | 0.20% | 3.00% | −0.10% | 0.10% | 0.20% | 1 | 1 | 1 | 3.10% | 0 | 1.20% |
| Max | 0.23 | 0.41 | 0.14 | 0.15 | 0.05 | 0.07 | 0.01 | 0.2 | 0.02 | 2.68 | 0.26 | 0.05 | 1.21 | 1 | 1 | 1 | 0.4 | 0.05 | 0.16 |
| Min | −0.2 | −0.36 | 0 | −0.11 | −0.04 | −0.1 | −0.02 | −0.22 | −0.04 | −4.99 | −0.27 | −0.06 | −0.44 | 0 | 0 | 0 | −0.27 | −0.06 | −0.23 |
| Std. dev. | 5.70% | 8.70% | 0.90% | 3.20% | 1.50% | 1.50% | 0.30% | 4.80% | 0.70% | 0.99 | 6.10% | 1.70% | 11.10% | 43.60% | 43.60% | 49.40% | 13.90% | 1.70% | 4.50% |
| Skewness | 0.26 | 0.33 | 10.8 | 0.37 | −0.75 | −0.45 | 0 | 0.24 | −1.34 | −1.31 | −0.3 | −0.17 | 2.58 | −1.13 | −1.32 | −0.32 | 0.09 | −0.25 | −0.71 |
| Kurtosis | 4.34 | 6.81 | 147.48 | 5.4 | 3.46 | 10.99 | 7 | 5.24 | 8.69 | 6.76 | 5.58 | 3.16 | 33.42 | 2.28 | 2.74 | 1.11 | 2.51 | 3.24 | 5.58 |
| Persistence | −0.01 | 0.03 | 0.53 * | 0.05 | 0.95 * | −0.03 | 0.67 | 0.03 | 0.36 * | 0.66 * | −0.01 | 0.30 * | 0.04 | 0.69 * | 0.77 * | 0.09 | 0.01 | 0.82 * | 0.06 |
| Obs. | 481 | 481 | 481 | 481 | 481 | 481 | 481 | 481 | 481 | 481 | 481 | 481 | 481 | 481 | 481 | 481 | 481 | 481 | 481 |
| Panel B: Correlation Matrix | | | | | | | | | | | | | | | | | | |
| DY | 1 | | | | | | | | | | | | | | | | | | |
| EP | 0.6 | 1 | | | | | | | | | | | | | | | | | |
| SVAR | 0.26 | 0.18 | 1 | | | | | | | | | | | | | | | | |

Bold

| | | | | | | | | | | | | | | | | | | | |
|---|---|---|---|---|---|---|---|---|---|---|---|---|---|---|---|---|---|---|---|
| LTR | 0 | 0.08 | 0.11 | 1 | | | | | | | | | | | | | | | |
| TMS | −0.05 | −0.13 | 0.04 | −0.05 | 1 | | | | | | | | | | | | | | |
| DFR | −0.21 | −0.23 | −0.14 | −0.44 | 0.13 | 1 | | | | | | | | | | | | | |
| INFL | 0.11 | 0.11 | −0.13 | −0.06 | −0.42 | −0.09 | 1 | | | | | | | | | | | | |
| CLAIMS | 0.1 | 0.15 | 0.14 | 0.08 | −0.22 | −0.05 | 0.12 | 1 | | | | | | | | | | | |
| INDP | −0.03 | −0.01 | −0.11 | −0.12 | 0.1 | 0.1 | −0.01 | −0.18 | 1 | | | | | | | | | | |
| CFNAI | −0.09 | −0.08 | −0.24 | 0.01 | 0.2 | 0.06 | −0.09 | −0.2 | 0.39 | 1 | | | | | | | | | |
| PERMIT | −0.06 | −0.15 | −0.16 | −0.08 | 0.22 | 0.12 | −0.11 | −0.2 | 0.14 | 0.21 | 1 | | | | | | | | |
| FEX | 0.16 | 0.15 | 0.08 | −0.15 | −0.07 | −0.07 | 0.04 | −0.03 | 0.01 | −0.06 | 0.02 | 1 | | | | | | | |
| OIL | −0.27 | −0.23 | −0.18 | −0.12 | −0.05 | 0.12 | 0.03 | 0.02 | 0.04 | 0.05 | 0.06 | −0.14 | 1 | | | | | | |
| MA | −0.31 | −0.17 | −0.25 | −0.08 | −0.06 | 0.06 | 0.06 | −0.14 | 0.21 | 0.19 | 0.04 | 0.01 | 0.15 | 1 | | | | | |
| MOM | −0.16 | −0.01 | −0.17 | −0.07 | −0.11 | −0.01 | 0.07 | −0.07 | 0.26 | 0.18 | 0.01 | 0.08 | 0.05 | 0.58 | 1 | | | | |
| VOL | −0.72 | −0.48 | −0.14 | −0.01 | 0.04 | 0.13 | −0.08 | −0.06 | 0 | 0.06 | 0.01 | −0.14 | 0.21 | 0.27 | 0.16 | 1 | | | |
| RSI | −0.85 | −0.55 | −0.12 | 0.02 | 0.04 | 0.15 | −0.07 | −0.05 | 0.02 | 0.06 | 0.03 | −0.13 | 0.25 | 0.36 | 0.21 | 0.8 | 1 | | |
| RS | −0.11 | −0.03 | −0.04 | −0.12 | −0.09 | −0.02 | 0.29 | 0.09 | −0.04 | −0.07 | −0.09 | −0.07 | 0.21 | 0.32 | 0.27 | 0.13 | 0.17 | 1 | |
| MARKET | −0.63 | −0.45 | −0.33 | 0.11 | 0.03 | 0.25 | −0.04 | −0.17 | −0.04 | 0.13 | 0.07 | −0.06 | 0.04 | 0.22 | 0.09 | 0.47 | 0.57 | −0.16 | 1 |

Notes. Panel A provides summary statistics for the predictive variables during the period from January 1974 to December 2013 covering 480 months. Panel B provides the correlation matrix for the predictive variables during the period from January 1974 to December 2013. The fundamental variables (DY, EP) and the technical variables are sector specific. For brevity, Table 3 only presents the fundamental and technical variables for the Oil and Gas sector. * indicates values significantly different from 0 at the 1% level.

## 4. Empirical Results for Sector Return Predictions

To analyze the individual predictive power of different variables in forecasting monthly sector returns, we start by computing univariate predictive regressions. We first analyze the predictive power of individual variables in sample and then focus on an out-of-sample analysis.

### 4.1. In-Sample Predictive Power of Individual Variables

For each of the predictive variables described in Section 3, we run the following univariate predictive regression individually for the full period (January 1974 to December 2013).

$$r_t = \alpha + \beta \ X_{t-1} + \varepsilon_t \tag{4}$$

where $r_t$ refers to the continuous monthly returns, and $X_{t-1}$ are the lagged indicators. In sample predictability tests are coefficient tests (H0: $\beta_i = 0$), testing the hypothesis that a specific variable (i) significantly predicts future returns. However, if time-series characteristics such as heteroscedasticity and persistence in predictive variables (Ferson et al. 2003) are not accounted for, in-sample results might be biased. We account for these effects by computing robust (Newey–West) standard errors.

For each predictive variable, Table 4 presents the regression coefficient and the statistical significance level inferred from robust t-statistics. Due to the large unpredictable component in stock returns, the $R^2$ statistics are small and do not exceed 1% in most cases. However, a monthly $R^2$ of only 0.5% can already result in substantial performance gains for investors and therefore represents an economically relevant level of return predictability (Kandel and Stambaugh 1996; Campbell and Thompson 2008). In addition, Table A1 in the Appendix shows the results of the in-sample univariate full sample univariate predictive regression using the reduced bias estimation method of Amihud and Hurvich (2009). Overall, the results in Table 4 are very similar to those obtained using the reduced bias estimation method.

**Table 4.** In-sample analysis of univariate predictive power (full sample univariate predictive regression).

| | Market | Oil and Gas | Manufacturing | Con Gds and Srvcs | Health Care | Tech and Tele | Financials |
|---|---|---|---|---|---|---|---|
| Fundamental Variables and Interest-Rate-Related Variables | | | | | | | |
| DY | −0.056 | 0.007 | −0.078 | −0.118 *** | −0.016 | −0.017 | −0.103 ** |
| EP | −0.059 | 0.010 | −0.061 | −0.103 ** | −0.006 | −0.015 | −0.099 ** |
| SVAR | −1.27 *** | −0.751 ** | −0.918 * | −0.985 ** | −0.711 | −1.472 ** | −1.016 |
| LTR | 0.105 * | 0.008 | 0.112 | 0.141 * | 0.124 * | 0.086 | 0.176 ** |
| TMS | 0.207 | 0.129 | 0.181 | 0.28 * | 0.069 | 0.298 * | 0.163 |
| DFR | 0.346 | 0.436 * | 0.369 | 0.401 * | 0.347 ** | 0.175 | 0.330 |
| Macroeconomic Variables | | | | | | | |
| INFL | −0.150 | −0.870 | 0.023 | −0.243 | −0.270 | −0.698 | 1.175 |
| CLAIMS | −0.087 * | −0.038 | −0.107 ** | −0.125 ** | 0.022 | −0.122 ** | −0.072 |
| INDP | 0.558 | 1.195 ** | 0.687 | 0.009 | 0.351 | 0.498 | 0.489 |
| CFNAI | 0.007 * | 0.006 * | 0.007 * | 0.006 | 0.002 | 0.009 ** | 0.009 ** |
| PERMIT | 0.051 | 0.042 | 0.061 * | 0.072 * | 0.015 | 0.043 | 0.078 * |
| FEX | −0.365 ** | −0.523 *** | −0.398 *** | −0.157 | −0.315 ** | −0.419 ** | −26.70% |
| OIL | −0.032 | −0.005 | −0.014 | −0.05 ** | −0.027 | −0.056 * | −0.023 |
| Technical Indicators | | | | | | | |
| MA | 0.009 | 0.001 | 0.010 | 0.004 | 0.002 | 0.009 | 0.006 |
| MOM | 0.009 | 0.002 | 0.005 | 0.005 | 0.008 | 0.009 | 0.006 |
| VOL | 0.004 | 0.004 | 0.006 | 0.012 ** | 0.001 | 0.001 | 0.013 ** |
| RSI | 0.007 | −0.016 | 0.008 | 0.032 * | 0.001 | −0.002 | 0.033 * |
| RS | - | −0.332 ** | −0.436 | −0.426 | −0.172 | 0.617 * | 0.122 |
| MARKET | 0.065 | 0.041 | 0.063 | 0.123 ** | 0.011 | 0.028 | 0.126 |

Notes. This table provides the regression coefficients and significance levels of in-sample univariate predictive regressions over the full period from January 1974 to December 2013 for the six industries and the market. We analyze 19 predictive variables, including fundamental variables and interest-rate-related variables, macroeconomic variables, and technical indicators. *, **, and *** indicate values significantly different from 0 at the 10%, 5%, and 1% level, respectively.

Analyzing fundamental and interest-rate-related variables, we find that the valuation ratio dividend-yield and the earnings-to-price ratio only have significant predictive power for the Financials and the Consumer Goods and Services sectors.

The fixed income factors, the long-term bond return (LTR), the term spread (TMS), and the default rate (DFR) all positively affect future stock market and industry returns. This supports earlier studies (Goyal and Welch 2008). However, the predictability of interest-rate-related data is rather low in that we find some statistically significant effects of fixed income variables only for the Consumer Goods and Services, the Financials, and the Health Care sectors.

In the group of fundamental variables, the stock variance has the largest predictive power for the overall stock market and industry returns. Negative coefficients indicate that an increase in the stock variance has a negative effect on future returns. This is in line with the results of earlier studies. For instance, Giot (2005) finds a negative and statistically significant relationship between stock index returns and the corresponding implied volatility indices.

In the group of macroeconomic variables, we observe that the initial claims of unemployment, the trade-weighted dollar index, and the Chicago Fed National Activity Index (CFNAI) have statistically significant power to predict the overall stock market excess returns. They are also the most important factors for predicting industry returns. The change in the initial claims of unemployment negatively affects stock returns, which is economically sensible, as higher unemployment claims indicate a stagnating or

contracting economic phase, typically with lower earnings. The trade-weighted dollar index negatively affects future industry returns. A higher dollar index reflects an appreciation of the dollar relative to a set of reference currencies, leading to higher prices for U.S. exports, resulting in a lower demand and lower expected revenues and subsequently in declining stock prices. All coefficients for the Chicago Fed National Activity Index (CFNAI) are positive, which is consistent with the objective of the CFNAI to signal growth above (below) the trend in the economy by positive (negative) index values. Building permits significantly predict the Manufacturing, the Consumer Goods and Services, and the Financials sectors. This supports the notion that the housing market is an early indicator for the state of the economy. The industry production index forecasts returns of the Manufacturing and Oil and Gas sectors at an economically significant level ($R^2$ strongly exceeding the 0.5% benchmark). As expected, oil price increases negatively affect future stock market and industry returns. However, this is statistically significant only for the Consumer Goods and Services and the Technology and Telecommunication sectors. This is in line with the results of Driesprong et al. (2008), who find that an increase in oil price tends to have a negative effect on energy-importing countries but a positive effect on energy-exporting countries. Therefore, the negative relation between oil price and stock market returns may have weakened in the U.S. over the last decade due to lowering energy import rates.

For the group of technical indicators, we find only little predictive power, which—at a first glance—seems contradictory to the results of Neely et al. (2014). However, a possible explanation is that we analyze a shorter period starting only in 1973, while Neely et al. (2014) begin in 1951. Most likely, financial markets have become more efficient over the last 60 years, limiting the predictive power of technical indicators. Moreover, since our sample size is substantially smaller, the statistical power of detecting predictability based on predictive regressions is lower. However, we find statistically significant and economically relevant ($R^2$ exceeding 0.5%) predictive power for at least one technical indicator for each industry. Therefore, including technical indicators might improve forecasts based on multivariate models.

### 4.2. Out-of-Sample Predictive Power of Individual Variables

The in-sample analysis of individual predictive variables provided an indication of which variables have predictive power to forecast industry returns and how changes in predictive variables affect future returns. Significant in-sample predictability, however, does not imply that investors can exploit return predictions to generate a superior portfolio performance. For instance, the relation between a predictive variable and future returns may not be stable over time and therefore cannot be used to improve performance. Therefore, a more relevant measure of return predictability for investors has to be based on an out-of-sample analysis (Neely et al. 2014).

To compute out-of-sample forecasts, we divide the full sample into an initial estimation sample and an out-of-sample evaluation sample. For each of the predictive variables described in Section 3, we run an univariate predictive regression (Equation 4) individually.[12]

The initial estimation sample ranges over 120 months (10 years). Additionally, we employ a 60-month holdout period to evaluate different forecast models on an ex ante basis. Therefore, our out-of-sample evaluation period ranges from January 1989 to December 2013. We compute 1-month-ahead out-of-sample forecasts recursively by re-estimating the forecast models for each month of the out-of-sample evaluation period using an expanding estimation window. To avoid any forward-looking bias, we only use data available until month (t) to compute forecasts for the subsequent month (t + 1). The out-of-sample predictability test builds on the mean squared forecast error (MSFE) of a prediction model (i) compared to the MSFE of the historical cumulative average (HA) forecast (Campbell and Thompson 2008). The historical average (HA) forecast is a very stringent out-of-sample benchmark, and most forecasts based on univariate predictive

regressions typically fail to outperform the historical average (Goyal and Welch 2003, 2008). The Campbell and Thompson (2008) out-of-sample R² ( $R_{OS}^2$ ) measures the proportional reduction in MSFE of the predictive regression forecast relative to the historical average. We compute $R_{OS}^2$ according to Equation (5):

$$R_{OS}^2 = 1 - \frac{MSFE(\hat{r}_t)}{MSFE(\bar{r}_t)} = 1 - \frac{\sum_{t=T_1+1}^{T}(r_t - \hat{r}_t)^2}{\sum_{t=T_1+1}^{T}(r_t - \bar{r}_t)^2},$$ (5)

where $r_t$ is the actual return, $\hat{r}_t$ is the forecast of the prediction model, and $\bar{r}_t$ is the historical cumulative mean return.

A positive $R_{OS}^2$ indicates that the predictive regression forecast exhibits a lower MSFE than the historical average. Monthly $R_{OS}^2$ are generally small due to the large unpredictable components in stock returns. However, a monthly $R_{OS}^2$ of 0.5% is economically significant, adding value for investors (Campbell and Thompson 2008). To test the statistical significance of $R_{OS}^2$, we compute the Clark and West (2007) MSFE-adjusted statistic, which allows for the testing of significant differences in prediction errors between the historical average forecast and a prediction model. It accounts for the usually higher level of noise in return forecasts compared to the historical cumulative average. [13] In Table 5, we present the results for the out-of-sample analysis.

**Table 5.** Out-of-sample analysis of univariate predictive power (monthly univariate predictive regressions).

| | Market | | Oil and Gas | | Manufacturing | | Con Gds and Srvcs | | Health Care | | Tech and Tele | | Financials | |
|---|---|---|---|---|---|---|---|---|---|---|---|---|---|---|
| | Ros² | t-Stat | Ros² | t-Stat | Ros² | t-Stat | Ros² | t-Stat | Ros² | t-Stat | Ros² | t-Stat | Ros² | t-Stat |
| Fundamental and Interest-Rate-Related Variables | | | | | | | | | | | | | | |
| DY | −0.07% | 0.18 | −0.42% | −0.87 | 0.48% | 0.78 | 0.24% | 1.41 * | −0.19% | −0.84 | −0.91% | −3.07 | 0.07% | 1.04 |
| EP | 0.03% | 0.45 | −0.56% | −1.25 | 0.29% | 0.75 | 0.24% | 1.26 | −0.20% | −0.95 | −0.83% | −1.55 | 0.16% | 1.03 |
| SVAR | 1.69% | 1.27 | −3.65% | 1.58 * | 0.44% | 0.87 | 0.33% | 0.81 | 0.12% | 0.41 | 1.48% | 1.24 | −0.59% | 0.87 |
| LTR | −0.43% | 0.33 | −0.53% | −1.8 | −0.69% | 0.16 | −0.74% | 0.52 | 0.29% | 0.86 | −0.15% | 0.01 | −0.62% | 0.46 |
| TMS | −1.06% | −0.13 | −0.85% | −0.88 | −1.08% | −0.38 | −0.74% | 0.44 | −1.01% | −1.42 | −0.54% | 0.22 | −0.70% | −0.2 |
| DFR | −1.61% | 0.16 | 0.11% | 0.81 | −0.80% | 0.38 | −1.80% | 0.42 | −1.58% | 0.45 | −1.16% | −1.42 | −2.04% | −0.18 |
| Macroeconomic Variables | | | | | | | | | | | | | | |
| INFL | −0.59% | −0.5 | −0.03% | 0.34 | −0.58% | −0.92 | −0.80% | −0.51 | −0.32% | −0.29 | −0.14% | 0.14 | −0.07% | 0.16 |
| CLAIMS | 1.17% | 2.62 *** | −0.21% | −0.36 | 1.70% | 2.77 *** | 2.17% | 3.11 *** | −0.44% | 0.04 | 0.99% | 2.60 *** | 0.04% | 0.42 |
| INDP | 0.61% | 0.78 | 3.54% | 1.80 ** | 1.09% | 0.92 | −1.58% | −0.79 | 0.09% | 0.36 | 0.07% | 0.39 | −0.44% | −0.26 |
| CFNAI | 3.06% | 2.09 ** | 1.82% | 1.90 ** | 2.87% | 1.85 ** | 1.89% | 1.93 ** | 0.01% | 0.19 | 2.56% | 2.38 *** | 3.22% | 2.06 ** |
| PERMIT | −0.43% | −0.2 | −0.44% | −0.44 | −0.35% | 0.15 | −1.26% | −0.38 | −0.31% | −1.14 | −0.29% | −0.62 | 0.30% | 0.88 |
| FEX | 1.88% | 1.95 ** | 2.82% | 2.41 | 2.05% | 1.97 ** | −0.24% | −0.35 | 1.46% | 1.77 ** | 1.31% | 1.91 ** | 0.15% | 0.55 |
| OIL | 0.39% | 0.95 | −0.34% | −1.22 | −0.33% | 0.03 | 1.63% | 1.79 ** | −0.73% | 0.4 | 0.94% | 1.33 * | −0.37% | 0.33 |
| Technical Indicators | | | | | | | | | | | | | | |
| MA | 0.55% | 0.87 | −0.54% | −1.83 | 0.79% | 0.96 | −0.61% | −0.69 | −0.41% | −2.26 | −0.08% | 0.38 | −0.28% | −0.36 |
| MOM | 0.82% | 1.16 | −0.41% | −1.18 | −0.16% | −0.11 | −0.24% | −0.4 | 0.78% | 1.44 * | −0.06% | 0.32 | −0.23% | 0.01 |
| VOL | −0.36% | −0.27 | −0.39% | −0.64 | 0.14% | 0.49 | 1.06% | 1.76 ** | −0.32% | −1.88 | −0.58% | −1.84 | 0.91% | 1.49 * |
| RSI | −0.20% | −0.81 | 1.17% | 1.75 ** | −0.19% | 0.5 | 0.55% | 1.12 | −0.46% | −0.51 | 0.58% | 0.81 | −1.22% | −0.45 |
| RS | NA | NA | −0.03% | 0.29 | −0.18% | −0.76 | 0.06% | 1.01 | −0.17% | −1.02 | −0.36% | −0.85 | 0.47% | 1.31 * |
| MARKET | 0.08% | 0.38 | −0.37% | −0.8 | 0.17% | 0.46 | 0.93% | 1.42 * | −0.38% | −0.89 | −0.46% | −1.86 | 0.83% | 1.07 |

Notes. This table provides an evaluation of the forecast errors of univariate predictive regression forecasts for 19 variables during the evaluation period from January 1989 to December 2013. The Ros² measures the percent reduction in MSFE for the forecast based on the predictive variable given in the first column relative to the historical average forecast. MSFE adjusted refers to the Clark and West (2007) MSFE-adjusted statistic for testing the null hypothesis that the MSFE of the historical average is less than or equal to the MSFE of the predictive regression. *, **, and *** indicate a statistically significant better forecast compared to the historical average at the 10%, 5%, and 1% level, respectively.

Like Goyal and Welch (2008), we find negative $R^2_{OS}$ statistics for the majority of fundamental and interest-related variables, indicating that the univariate predictive regression forecasts fail to outperform the historical average in terms of MSFE. For some predictive variables (e.g., SVAR for Oil and Gas), the $R^2_{OS}$ statistic is negative, while at the same time, the MSFE-adjusted t-statistic is positive. This is a well-known phenomenon and stems from the fact that the MSFE-adjusted statistic accounts for the higher volatility in predictive regression forecasts compared to the historical average (Clark and West 2007; Neely et al. 2014). The most promising predictive variables are the initial claims of unemployment, the trade-weighted dollar index, and the Chicago Fed National Activity Index. In addition, for some sectors, the production index, oil price, stock variance, and technical indicators significantly outperform the historical averages. Overall, the forecast power of individual predictive variables seems rather limited. We find only little predictive power for fundamental ratios and technical indicators, which—at a first glance—seems contradictory to the results of Neely et al. (2014). Possible reasons for this we already discussed above. However, we find statistically significant and economically relevant ($R^2$ exceeding 0.5%) predictive power for at least one technical indicator for each sector. Therefore, combining several different predictive variables in multivariate models and including technical indicators might enhance the forecast performance. We discuss the multivariate models in the next section.

### 4.3. Multivariate Prediction Models

Next, we explore the predictive power of various competing multivariate prediction models. We expect that the predictive power of a forecast model increases when jointly employing several predictive variables in a multivariate model. However, employing all 19 variables simultaneously in an OLS model likely generates poor out-of-sample forecasts for three reasons: First, including all variables probably results in a relatively high level of estimation error due to the large number of coefficient estimates. Second, different predictive variables might partly capture the same underlying information, and the potential correlations between individual predictive variables might lead to unstable and biased coefficient estimates. Third, not all predictive variables might be relevant for all sectors. While it is possible to draw inferences based on economic theory to decide which predictive variables should be most relevant for a specific sector, some relations might be less obvious. Simply picking the best variables for each sector based on the forecast-ability of the entire sample incorporates a look-ahead bias and therefore is not adequate for our out-of-sample ex ante approach. Consequently, for each sector, we include all variables in the forecast model.

To circumvent potential problems when using OLS, we employ alternative multivariate approaches and compare their performance. All multivariate prediction models are computed recursively based on data available until the month (t) to compute forecasts for the subsequent month (t + 1) for each month of the out-of-sample evaluation period using an expanding estimation window. As a robustness check, we also compute forecasts based on a 120-month rolling estimation window and obtain very similar results. One could argue that different regimes over the sample period require a regime shift model for stock prediction. The problem with these models is the correct identification of the underlying regimes. Hammerschmidt and Lohre (2018) use regime shift models and do not find a superior performance of these models compared to that of a simple rolling window approach. We include OLS with all predictors as a benchmark model. The remainder of this section presents the employed multivariate prediction models.

### 4.3.1. Model Selection Based on Information Criteria

The first approach for selecting the best model from a large set of potential predictive variables builds on information criteria. As in Pesaran and Timmermann (1995), Bossaerts and Hillion (1999), and Rapach and Zhou (2013), we let the Schwarz information criterion (SIC) decide on the best model in that we allow up to three predictors in any combination

in the model. At each point in time, we calculate the Schwarz's Bayesian information criterion (SIC) for each of the alternative models and select the model with the smallest information criterion for forecasting the next period. This approach simulates the investor's search for a forecasting model by applying standard statistical criteria for model selection. Different variables might be selected for different sectors, and the variables included in the forecast model might vary over time due to re-selecting the optimal combination of variables based on the SIC in each month.

### 4.3.2. Predictive Regression via the LASSO

Tibshirani (1996) develops a least absolute shrinkage and selection operator (LASSO) designed for estimating models with numerous regressors. LASSO is a regularization technique, which performs linear regression with a penalty term that constrains the size of the estimated coefficients. It generally shrinks OLS regression coefficients towards zero, and some coefficients are shrunk exactly to zero (Tibshirani 1996). Therefore, the LASSO performs variable selection, alleviates the problem of coefficient inflation due to multi-collinearity, and produces shrinkage estimates with potentially higher predictive power than ordinary least squares. The LASSO estimates are defined by

$$(\hat{\alpha}, \hat{\beta}) = argmin \left\{ \sum_{t=1}^{T} (r_t - \alpha - X_{t-1}'\beta)^2 + \lambda \sum_{k=1}^{K} |\beta_k| \right\} \tag{6}$$

where $\lambda$ is the regularization parameter, T is the number of observations, $X_{t-1}$ contains the predictive variables, and $r_t$ is the return in the subsequent period (the prediction target). To compute sector return forecasts, we run a predictive regression based on the LASSO coefficients for each sector and in each month.

### 4.3.3. Predictive Regressions Based on Principal Components

The third approach is a prediction based on principal components. The basic idea is to reduce the large set of potential predictive variables and to extract a set of uncorrelated latent factors (principal components) that capture the common information (co-movements) of the predictors. Model complexity is thereby reduced, and noise in the predictive variables is filtered out, reducing the risk of sample overfitting (Ludvigson and Ng 2007; Neely et al. 2014; Hammerschmidt and Lohre 2018). We compute the principal component forecast based on a predictive regression on these first principal components:

$$r_{t+1} = \hat{\alpha}_{\square} + \sum_{k=1}^{K} \hat{\beta}_k F_{k,t}, \tag{7}$$

where $F_{k,t}$ is the vector containing the kth principal component. For the U.S. stock market, Neely et al. (2014) report that the principal components forecast based on fundamental or technical variables significantly outperforms the historical average forecast. We compute the first principal components based on standardized predictor variables. A critical issue is the selection of the optimal number of principal components to include in predictive regressions. We let the Schwarz information criterion (SIC) decide the optimal number of principal components at each point in time and for each sector.

### 4.3.4. Target-Relevant Factors

Principal components identify latent factors with the objective of explaining the maximum variability within predictors. A potential drawback of this approach is that it ignores the relationship between the predictive variables and the forecast target. Partial least squares regression (PLS), pioneered by Wold (1975), aims to identify latent factors that explain the maximum variation in the target variable. Kelly and Pruitt (2013, 2015) extend PLS to what they call a three-pass regression filter for estimating target-relevant factors (TRFs). In a simulation study and two empirical applications, Kelly and Pruitt

(2015) provide evidence that the TRF filter achieves a high forecast performance, outperforming competing approaches such as principal components. We employ the TRF filter to extract one target-relevant factor from the set of 19 predictive variables for each sector in each month. Therefore, the target-relevant factor may include different predictive variables over time and for each sector. To compute sector return forecasts, we run a predictive regression on the target-relevant factor for each sector in each month.

### 4.3.5. Forecast Combinations

Bates and Granger (1969) report that combining forecasts across different prediction models often generates forecasts superior to all individual forecasts. If individual forecasts are not perfectly correlated, i.e., different predictive variables capture different information on the overall economy or sector conditions, the combined forecasts are less volatile and usually have lower forecast errors (Hendry and Clements 2004; Timmermann 2006). An intuitive way of using the predictive power of several predictive variables is combining the forecasts of the individual univariate predictive regressions. The simplest way to merge individual forecasts is simple averaging, which means that each forecast obtains a weight of $\omega = 1/K$, where K is the number of forecasts.

Bates and Granger (1969) propose choosing forecast weights that minimize the historical mean squared forecast error (optimal combination). However, a number of empirical applications suggest that this optimal combination approach usually does not achieve a better forecast compared to the simple average of individual forecasts (Clemen 1989; Stock and Watson 2004).[14] Stock and Watson (2004) and Rapach et al. (2010) employ mean squared forecast error (MSFE)-weighted combination forecasts, which weigh individual forecasts based on their forecasting performance during a holdout out-of-sample period.[15] Rapach et al. (2010) find that simple and MSFE-based weighted combination forecasts outperform the historical average in predicting the U.S. equity market risk premium. We employ MSFE-weighted combinations of univariate predictive regressions to forecast sector returns.

Additionally, we employ a variable selection process, ensuring that the combination forecasts contain only relevant variables. The variable selection process depends on the ability of a predictive variable to forecast returns at significant margins. Variables are selected on an ex ante basis. That is, for the decision as to whether a specific variable is included in forecasting the return for the subsequent period (t + 1), we rely on the univariate predictive regression including returns until the current period [0, t]. A variable is only included in the combination forecast if its regression coefficient estimate is significant at the 10% level, using robust (Newey–West) standard errors. We then use all significant variables to compute forecasts based on univariate predictive regressions, which we subsequently pool to obtain an MSFE-weighted combination. While p-values should generally only be used to retain or reject a stated hypothesis, it is a common practice to misuse p-values for selecting relevant variables. Therefore, we include this approach into our empirical analysis.

Forecasts are expected to improve not only when combining forecasts of univariate regressions based on different predictive variables but also when combining forecasts of different forecast models. In this spirit, we compute a consensus forecast that simply combines all aforementioned forecast models by taking the simple average of all forecasts.

### 4.4. Out-of-Sample Predictive Power of Multivariate Models

#### 4.4.1. Prediction Quality

Combining the predictive power of fundamental, macroeconomic, and technical variables should result in superior forecasts. In Table 6, we present the findings for the multivariate forecast models in which all variables are included. The employed forecast model is displayed in the first column of Table 6. Comparing the forecast performance for different sectors, it is evident that the returns of some sectors (e.g., Oil and Gas) are more

predictable than the returns of others (e.g., Health Care). For the Oil and Gas (Consumer Goods and Services) sector, seven (six) out of eight prediction models significantly outperform the historical average in forecasting future returns, as indicated by the significant t-statistic at the 10% level. For the Manufacturing, the Technology and Telecommunication, and the Financials sectors, the target-relevant factor (TRF) approach and the MSFE-weighted forecast combination model (FC-MSFE-w.avrg) significantly outperform the historical average. For the Health Care sector, no prediction model outperforms the historical average at significant levels (for the reasons mentioned above). However, the vast majority of multivariate prediction models achieve positive MSFE-adjusted t-statistics, indicating an outperformance compared to the historical average after controlling for noise in forecasts.

**Table 6.** Multivariate forecasts with all predictive variables: Forecast evaluation based on MSFE.

| Model | Market | | Oil and Gas | | Manufacturing | | Con Gds and Srvcs | | Health Care | | Tech and Tele | | Financials | |
|---|---|---|---|---|---|---|---|---|---|---|---|---|---|---|
| | $R_{OS}^2$ | t-Stat | $R_{OS}^2$ | t-Stat | $R_{OS}^2$ | t-Stat | $R_{OS}^2$ | t-Stat | $R_{OS}^2$ | t-Stat | $R_{OS}^2$ | t-Stat | $R_{OS}^2$ | t-Stat |
| OLS | −4.07% | 1.10 | −1.32% | 1.88 ** | −3.62% | 1.17 | −6.35% | 1.77 ** | −8.33% | 1.17 | −1.85% | 1.17 | −10.09% | 0.69 |
| SIC−3 | −4.91% | −1.62 | 0.29% | 0.79 | −1.36% | 0.09 | −3.26% | −0.22 | −4.50% | −1.40 | 0.18% | 1.08 | −4.37% | 0.56 |
| LASSO | 0.48% | 0.94 | 1.35% | 1.64 * | 1.25% | 0.91 | −0.71% | 0.85 | −1.36% | 0.74 | 1.08% | 1.18 | −0.50% | 0.69 |
| Principal components | 0.37% | 0.80 | 1.36% | 1.30 * | 0.43% | 0.73 | 1.41% | 1.94 ** | −0.05% | −0.09 | −0.05% | 0.48 | 0.55% | 0.90 |
| TRF | 3.70% | 1.90 ** | 3.23% | 2.25 ** | 3.38% | 1.79 ** | 0.94% | 2.24 ** | −5.54% | 0.80 | 2.56% | 1.71 ** | 1.64% | 1.43 * |
| FC MSFE w.avrg. | 0.85% | 1.70 ** | 0.82% | 2.04 ** | 0.80% | 1.55 * | 1.00% | 2.19 ** | 0.21% | 0.93 | 0.54% | 1.58 * | 0.88% | 1.28 * |
| FC VS-MSFE w.avrg. | 0.56% | 0.95 | 2.34% | 2.00 ** | 0.18% | 0.44 | 0.90% | 1.47 * | −0.91% | 0.12 | 0.88% | 1.21 | 1.40% | 1.20 |
| Consensus (avrg) | 1.93% | 1.29 * | 3.79% | 1.85 ** | 2.04% | 1.14 | 1.68% | 1.73 ** | −0.29% | 0.76 | 1.77% | 1.37* | 1.00% | 0.90 |

Notes. This table provides an evaluation of the forecast error of multivariate forecast models including all 19 predictive variables. The evaluation period is from January 1989 to December 2013. The $R_{OS}^2$ measures the percent reduction in MSFE for the multivariate forecast model given in the first column relative to the historical average forecast. MSFE adjusted refers to the Clark and West (2007) MSFE-adjusted statistic for testing the null hypothesis that the MSFE of the historical average is less than or equal to the MSFE of the predictive regression. Multivariate forecast models: OLS refers to the ordinary least squares regression model, SIC-3 reflects the multivariate regression model with variable selection based on the Schwartz information criterion (SIC), LASSO is the least absolute shrinkage and selection operator, principal components is the principal component regression model, TRF shows the target-relevant factor approach, FC-MSFE-w.avrg. is the forecast combination model combining univariate predictive regression forecasts with all indicators and forecast outputs weighted with the historical mean squared forecast error (MSFE), FC-VS-MSFE-w.avrg. also reflects the forecast combination of univariate predictive regressions but including only significant indicators at the 10% level, and the consensus forecast is the simple average of all aforementioned forecast models. *, and **indicate statistically significantly better forecasts compared to the historical average at the 10% and 5% level, respectively.

Comparing the forecast accuracy of different prediction models, we find that the best forecast models are the target-relevant factor approach (TRF) and the MSFE-weighted forecast combination model (FC-MSFE-w.avrg). For the overall stock market and for five out of six sectors, they generate statistically significantly superior forecasts compared to the cumulative historical average. The forecasts are also economically valuable, indicated by the $R_{OS}^2$ statistics above the 0.5% benchmark. Pre-selecting relevant variables before computing a forecast combination does not seem to add value relative to the forecast combination model that simply includes all variables. For all sectors, the variable selection model (FC-VS-MSFE-w.avrg) provides lower MSFE-adjusted statistics than the forecast combination model with all variables (FC-MSFE-w.avrg).

The regularization technique (LASSO), which shrinks coefficients towards zero to alleviate coefficient inflation and to perform variable selection, generates positive and economically significant $R_{OS}^2$ statistics (above the 0.5% benchmark) for three out of six sectors. The MSFE-adjusted t-statistic is positive for all sectors and the market, indicating that the LASSO forecast outperforms the historical average for all sectors. However, this

effect is only statistically significant for the Oil and Gas sector. The results of the principal component (PC) forecast are similar to those of the LASSO. Positive MSFE-adjusted statistics indicate that the PC forecasts outperform the historical average (after controlling for noise) for four out of six sectors and the market. However, the outperformance is only statistically significant for two sectors (Oil and Gas and Consumer Goods and Services).

The consensus forecast, which is a simple average of all multivariate models, is also very promising. It generates economically significant forecasts for the market index and all sectors except for the Health Care sector. For the Oil and Gas, the Consumer Goods and Services, and the Technology and Telecommunication sectors, the consensus forecast outperforms the historical average at statistically significant levels.

The simple multivariate OLS model and the model selection based on the Schwarz information criteria (SIC) generate the noisiest estimates, with negative $R^2_{OS}$ statistics for most sectors as well as for the overall stock market index. However, the MSFE-adjusted t-statistic, which controls for higher noise in the return prediction, is positive for most sectors, indicating that the forecasts are superior to the historical average forecast in most cases and may add economic value when included in asset allocation decisions. OLS outperforms the historical average at statistically significant levels for two of the six sectors.

### 4.4.2. Superior Predictive Ability Test

When analyzing the performance of different potential forecast models, data snooping is a natural concern. To control for data snooping, we apply Hansen's (2005) test for superior predictive ability (SPA test). The SPA test allows for comprehensive testing of several forecast models and ensures that the results are robust against biases from data snooping. The SPA test builds on White's (2000) 'reality check' (RC) but is more powerful and less sensitive to the inclusion of poor and irrelevant alternatives. Consequently, the SPA test detects whether there is at least one superior forecast model. However, it is not able to identify all superior strategies. Therefore, the SPA test tests the null hypothesis that the benchmark is not inferior to any alternative forecast model. In line with the previous analysis, we use the expanding historical average forecast as the benchmark model.

We employ a bootstrap implementation of the SPA test. The implementation is based on the stationary bootstrap with an average block size of 12 and 10,000 resamples. The predictive ability of each forecasting strategy is defined in terms of its expected loss ($L_k$). As a loss function, we use the absolute forecast error in line with Hansen (2005). The results of the SPA tests are presented in Table 7.

**Table 7.** Multivariate forecasts with all predictive variables: Hansen's (2005) SPA tests based on the MSFE loss function.

| Model | Market | Oil and Gas | Manufacturing | Con Gds and Srvcs | Health Care | Tech and Tele | Financials |
|---|---|---|---|---|---|---|---|
| OLS | 100.00% | 100.00% | 100.00% | 100.00% | 100.00% | 100.00% | 100.00% |
| SIC-3 | 100.00% | 100.00% | 100.00% | 100.00% | 100.00% | 29.52% | 100.00% |
| LASSO | 100.00% | 100.00% | 38.59% | 100.00% | 100.00% | 100.00% | 100.00% |
| Principal components | 45.47% | 13.05% | 18.72% | 33.89% | 15.26% | 27.95% | 37.46% |
| TRF | 20.60% | 16.39% | 8.35% | 48.61% | 100.00% | 4.66% | 44.64% |
| FC-MSFE-w.avrg. | 7.05% | 2.75% | 2.71% | 4.03% | 25.25% | 0.68% | 16.07% |
| FC-VS-MSFE-w.avrg. | 37.08% | 0.15% | 100.00% | 14.67% | 49.46% | 7.47% | 42.58% |
| Consensus (avrg) | 47.24% | 1.98% | 12.41% | 18.12% | 100.00% | 7.73% | 100.00% |

Notes: This table reports the results of Hansen's (2005) SPA test for multivariate forecast models compared to the historical average forecast in the out-of-sample period from January 1989 to December 2013 for a loss function based on mean squared forecast errors. Multivariate forecast models: OLS refers to the ordinary least squares regression model, SIC-3 reflects the multivariate regression model with variable selection based on the Schwartz information criterion (SIC), LASSO is the least absolute shrinkage and selection operator, Principal components is the principal component regression model, TRF shows the target-relevant factor approach, FC-MSFE-w.avrg. is the forecast combination model combining univariate predictive regression forecasts with all indicators and forecast outputs weighted with the historical mean squared forecast error (MSFE),

FC-VS-MSFE-w.avrg. also reflects forecast combination of univariate predictive regressions but including only significant indicators at the 10% level, and the consensus forecast is the simple average of all aforementioned forecast models. The results indicate that for four out of six sectors and for the market, we need to reject the null hypothesis, indicating that at least one forecast model outperforms the historical average even after controlling for data mining. For Health Care and Financials, we cannot reject the null hypothesis, suggesting that these sectors are more difficult to forecast or that we did not include the relevant indicators for these sectors in our prediction model. The result of the SPA test validate that the MSFE-weighted forecast combination model (FC-MSFE-w.avrg) is the best forecast model when evaluating the forecast models according to their forecast error.

## 5. Asset Allocation Strategies with Return Forecasts

Our analyses of the forecast performance based on the MSFE in the previous section provide promising insights that, in contrast to most literature and our own hypothesis, using well-specified prediction models may add value. The most important and essential question for investors, however, is whether using these return forecast models in asset allocation decisions results in higher risk-adjusted returns. Interestingly, Leitch and Tanner (1991) find only a weak relationship between MSFE and forecast profitability, and Cenesizoglu and Timmermann (2012) suggest that although return prediction models may produce higher out-of-sample mean squared forecast errors, they may still add economic value when used to guide portfolio decisions. Therefore, we analyze the benefits of multivariate forecast models for investors when used in asset allocation models.

### 5.1. Portfolio Optimization Procedure

We primarily rely on the BL model to compute optimal asset allocations based on return forecasts and compute traditional Markowitz (1952) and Bayes-Stein (Jorion 1985, 1986) portfolios as a robustness check. Our approach is as follows: We use the monthly sector-level return forecasts resulting from multivariate regression models to compute monthly BL-optimized portfolios. Then, we evaluate the portfolio performance of each forecast model compared to the historical average forecast and relative to the market portfolio as well as to a passive equally weighted (1/N) portfolio for the full sample and for sub-periods. DeMiguel et al. (2009) reports that the 1/N portfolio is a stringent benchmark, and many asset allocation models fail to outperform this naïve benchmark. One might argue that the long-term character of our asset allocation approach requires a dynamic portfolio strategy derived from optimizing a long-term objective function. However, Diris et al. (2015) show that the much simpler single-period portfolio optimization problem performs almost equally well for moderate levels of risk aversion. Therefore, we limit our analysis to single-period portfolio optimization strategies, with updates on a monthly basis when new information becomes available.

Following Campbell and Thompson (2008) and Neely et al. (2014), we estimate the covariance matrix based on a rolling estimation window of 60 months and use a risk-aversion coefficient of 5.[16] Although several methods exist to estimate asset variances and covariances more elaborately, we focus in this study on return forecast and employ a standard sample estimate for the covariance matrix. This approach is also supported by Lan (2014), who shows that for improving portfolio performance, it is more important to include a time-varying risk premium than time-varying volatility. We measure the reliability of each forecast $\Omega$ as the mean squared forecast error (MSFE) of each forecast using a 60-month rolling estimation window. Since we use the 1/N portfolio as benchmark to evaluate performance, the 1/N portfolio is also the reference portfolio used to compute 'implied' returns for the BL model. The parameter $\tau$, which controls for the level of deviation from the benchmark, is set to 0.1.[17]

*5.2. Performance Evaluation*

To evaluate the performance of sector-level forecasts in an asset allocation framework, we employ several performance measures. We compute the portfolio's average out-of-sample return and volatility, as well as the Sharpe ratio as the average excess return (average return minus risk-free rate) divided by the volatility of out-of-sample returns. We test if the difference in Sharpe ratios of two portfolios is statistically significant based on the significance test for Sharpe ratios proposed by Opdyke (2007).[18]

Following Campbell and Thompson (2008), Ferreira and Santa-Clara (2011), and Neely et al. (2014), among others, we measure the economic value of return forecasts based on the certainty equivalent return (CER). The CER gain is the difference between the CER for an investor who uses a predictive regression forecast for sector returns and the CER for the market portfolio. We use annualized portfolio returns and annualized return volatilities to compute the CER gain and interpret it as the annual portfolio management fee (in percentage points) that an investor would be willing to pay to have access to the predictive regression forecast instead of investing passively in the market portfolio. A drawback of the Sharpe ratio and the CER gain is that both measures only use portfolio returns and volatility, ignoring any higher moments. As an alternative, we calculate the Omega measure that is calculated as the ratio of average gains to average losses (Keating and Shadwick 2002), where gains (losses) are returns above (below) the risk-free rate.

To measure the trading volume necessary to implement asset allocation strategy i based on a return forecast model, we compute the portfolio turnover $PT_i$ for each strategy as the average absolute change in the portfolio weight $\omega$ over T rebalancing points in time and across N assets (DeMiguel et al. 2009):

$$PT_i = \frac{1}{T} \sum_{t=1}^{T} \sum_{j=1}^{N} \left( \left| \omega_{i,j,t+1} - \omega_{i,j,t+} \right| \right), \tag{8}$$

in which $\omega_{i,j,t}$ is the weight of asset j at time t in strategy i; $\omega_{i,j,t+}$ is the portfolio weight before rebalancing at t + 1; and $\omega_{i,j,t+1}$ is the desired portfolio weight at t + 1. $\omega_{i,j,t}$ is usually different from $\omega_{i,j,t+}$ due to changes in asset prices between t and t + 1.

The economic profitability of an active asset allocation strategy critically depends on transaction costs required to implement the strategy. Assuming a realistic level of transaction costs might be challenging because transaction costs typically differ for different investor types (e.g., private and institutional investors). Therefore, we follow Han (2006) and Kalotychou et al. (2014) and compute break-even transaction costs (BTCs). We define the BTC as the level of variable transaction costs for which the active investment strategy based on a return forecast model achieves the same certainty equivalent return (CER) as the market portfolio. Consequently, the active investment strategy based on a return forecast model is superior relative to a passive investment in the market portfolio as long as the variable transaction costs are below the BTC. Finally, we evaluate the risk-adjusted performance of the optimized sector portfolios using the Carhart four-factor model (Fama and French 1993; Carhart 1997).[19]

*5.3. Performance of Sector Return Predictions in Portfolio Optimization*

5.3.1. Risk-Adjusted Performance

We use the monthly sector-level return forecasts resulting from multivariate regression models to compute monthly BL-optimized portfolios. The portfolios consist of the six sector indices and the risk-free rate. To evaluate the contribution of return forecasts, we employ four benchmark portfolios: the market portfolio; a passive 1/N buy-and-hold portfolio, which equally weighs all six sector indices; and two BL-optimized portfolios, which use either the cumulative return average or a rolling return average for each sector as return forecasts.

Table 8 provides the performance measures for the four benchmark portfolios (row 1 to 4) and the asset allocations based on multivariate return forecast models (rows 5 to 12).

In addition, Figure A1 in the Appendix shows the distribution of monthly portfolio returns for all allocation strategies and Figure A2 shows the portfolio weights allocated to sectors and liquidity for each strategy. The first column of Table 8 displays the underlying return forecast model. We find that all asset allocations based on multivariate return forecast models outperform all benchmark portfolios, with higher Sharpe ratios, Omega measures, and certainty-equivalent ratios. All forecast models yield significantly larger Sharpe ratios than the market index, and all forecast models except for the simple OLS forecast significantly outperform the passive 1/N portfolio. This result was not expected, given our previous analysis of $R^2_{OS}$ statistics, which were not that compelling for all forecast models. A weak relation between the statistical out-of-sample forecast evaluation based on the MSFE (depicted in Table 6) and the economic value of a forecast model was already reported by Cenesizoglu and Timmermann (2012). Compared to the market and the 1/N portfolio, all allocations based on return forecast models enhance the portfolio return. Except for the OLS and the SIC forecast, asset allocations based on forecast models also reduce portfolio risk (volatility) compared to the passive benchmark portfolios.

**Table 8.** Analysis of sector forecasts in asset allocation: BL portfolios (unconstrained).

| Performance Measure | Return | Volatility | Sharpe | Omega | CER | CER-Gain | Turnover | BTC |
|---|---|---|---|---|---|---|---|---|
| Market | 10.39% | 15.14% | 0.46 | 1.41 | 0.05 | 0.00% | 0.00 | |
| 1/N | 11.21% | 14.47% | 0.54 | 1.49 | 0.06 | 1.32% | 0.00 | |
| Cumulative average | 12.32% | 12.41% | 0.72 *** | 1.72 | 0.08 | 3.82% | 2.11 | 1.81% |
| Moving average (60 m) | 13.79% | 13.54% | 0.77 *** | 1.80 | 0.09 | 4.55% | 2.92 | 1.55% |
| OLS | 16.68% | 16.79% | 0.79 * | 2.20 | 0.10 | 4.98% | 22.48 | 0.22% |
| SIC-3 | 14.79% | 14.49% | 0.79 ** | 1.98 | 0.10 | 4.88% | 7.27 | 0.67% |
| LASSO | 12.90% | 10.30% | 0.93 ***× | 2.03 | 0.10 | 5.59% | 6.79 | 0.82% |
| Principal components | 13.56% | 11.39% | 0.90 ***× | 1.91 | 0.10 | 5.66% | 3.71 | 1.53% |
| TRF | 16.92% | 12.33% | 1.10 ***×+ | 2.31 | 0.13 | 8.46% | 7.12 | 1.19% |
| FC MSFE w.avrg. | 12.80% | 11.93% | 0.79 ***× | 1.78 | 0.09 | 4.59% | 2.20 | 2.09% |
| FC VS MSFE w.avrg. | 13.11% | 11.48% | 0.85 ***× | 1.84 | 0.10 | 5.15% | 2.91 | 1.77% |
| Consensus (avrg) | 14.45% | 11.42% | 0.97 ***× | 2.06 | 0.11 | 6.54% | 7.11 | 0.92% |

Notes. This table reports the portfolio performance measures for an investor who allocates monthly between the six analyzed U.S. sectors based on the forecast model given in the first column. The asset allocation builds on the unconstrained Black and Litterman (1992) model. The investment universe includes the six sector indices. The first row shows the performance of the overall U.S. stock market index and the second row the performance of a passive equally weighted (1/N) sector portfolio as benchmark strategies. The evaluation period is from January 1989 to December 2013. 'Return' is the annualized time-series mean of monthly returns, and 'Vola' denotes the associated annualized standard deviation. 'Sharpe' shows the annualized Sharpe ratio and 'Omega' the Omega measures for each portfolio. 'CER' is the certainty equivalent return. 'CER-Gain' is the gain in CER for an investor who actively invests based on the indicated forecast model compared to an investor who passively invests in the 1/N portfolio. 'BTC' is the break-even transaction cost, which is the level of variable transaction costs per trade for which the active investment strategy based on the forecast model shown in the first column yields the same certainty equivalent return (CER) as the passive market portfolio. *, **, (×, ××), and [+, ++] indicate a statistically significantly higher Sharpe ratio compared to the market (the buy-and-hold 1/N portfolio) [the BL-optimized portfolio based on the cumulative historical average as a return forecast] at the 5% and 1% level, respectively.

In line with the analysis based on the MSFE in Table 6, we find that the best return forecast model is the target-relevant factor approach (TRF), followed by the consensus and the LASSO forecast, yielding the highest out-of-sample performance and outperforming the market and the passive 1/N portfolio at highly significant levels.

### 5.3.2. Gain in Certainty Equivalent Return (CER Gain)

The CER gain illustrates that before transaction costs, investors could pay a management fee as high as 846 (559) basis points per year and still benefit from active investing based on the TRF (LASSO) forecast model, rather than investing passively in the market portfolio. In line with Neely et al. (2014) and Hammerschmidt and Lohre (2018), we also find a significant outperformance for the principal component forecast model. In this case, investors could pay a fee of up to 566 basis points per year when actively investing based on the principal components forecast and still be better off than investing passively in the market portfolio. The outperformance for the MSFE-weighted average forecasts is lower. Investors would be willing to pay a performance fee of 459 basis points per year for the combination forecast including all variables and 515 basis points for the combination forecast, which conducts variable selection before computing an MSFE-weighted combination forecast.

### 5.3.3. Turnover and Break-Even Transaction Costs (BTCs)

The required turnover and associated transaction costs for the different allocation models (reported in column 7) vary substantially depending on the forecast model. This is associated with the different levels of noise included in the forecast models. The portfolio turnover is highest for the OLS model, followed by the SIC and the TRF forecasts, indicating that these models deliver the noisiest return predictions. The MSFE-weighted combination forecast results in the lowest turnover due to the more stable return forecasts over time. The optimal choice of a forecasting model depends also on the level of the transaction costs, which are necessary to implement the respective prediction-based asset allocation model. To circumvent problems associated with identifying an adequate level of transaction costs, we compute break-even transaction costs (BTCs).

The last column of Table 8 presents the BTCs for each forecast model. We observe for the MSFE-weighted average forecast that the variable transaction costs must not exceed 209 basis points so that the model still performs better compared to the market portfolio. For the principal components (the TRF) model, the asset allocation based on forecast models is preferable to the market portfolio as long as the variable transaction costs are lower than 153 (119) basis points. Kalotychou et al. (2014) approximate total trading costs to be seven basis points for institutional investors trading U.S.-sector ETFs. This is the sum of average bid–ask spreads for U.S. sector ETFs (5 bp) and market impact costs (2 bp). At this realistic level of transaction costs, all forecast models are beneficial to investors, and the performance ranking of the different return forecast models is the same as that shown in Table 8. Angel et al. (2011) estimate that transaction costs for individual U.S. large-cap stocks are about 40 basis points. This underpins the benefits of using ETFs for asset allocation. However, the BTCs suggest that all predictions models, except for the simple OLS forecast, would be beneficial compared to the market index, even if transaction costs were 40 basis points.

### 5.3.4. Superior Predictive Ability Test

To control for data snooping, we rerun Hansen's (2005) test for superior predictive ability (SPA test). Again, we use a bootstrap implementation of the SPA test with stationary bootstrap, an average block size of 12, and 10,000 resamples. Now, as a loss function, we employ the (negative) portfolio returns $LL_{i,t} = -r_{i,t,}$ and the (negative) risk-adjusted excess returns $LL_{i,t} = -\frac{r_{i,t,} - r_{f,t}}{\sigma_k}$ of the BL-optimized portfolios for the different forecast models (i) over the months (t) of the evaluation period. We run the SPA tests against four benchmark portfolios: the market portfolio; a passive 1/N buy-and-hold portfolio, which equally weighs all six sector indices; and two BL-optimized portfolios, which use either the cumulative return average or a rolling return average for each sector as return forecasts.

We present the results of the SPA tests in Table 9. Panel A presents the results when using the (negative) portfolio returns as loss function, whereas Panel B depicts the results for the (negative) risk-adjusted excess returns as loss function. The analysis based on returns (Panel A) shows that all forecast models except the LASSO beat the market and the 1/N portfolio at significant margins. However, only the principal component, TRF, and MSFE-weighted combination forecast outperform the cumulative historical average, and no forecast model beats the moving average in terms of returns.

**Table 9.** SPA tests based on returns and risk-adjusted returns.

| | Panel A | | | Panel B | | |
| | Test on Returns | | | Test on Risk-Adjusted Returns | | |
| | Test vs. Market | Test vs. 1/N | Test vs. Cum. Average | Test vs. Market | Test vs. 1/N | Test vs. Cum. Average |
|---|---|---|---|---|---|---|
| Cumulative avergare | 4.36% | 8.06% | | 4.45% | 8.63% | |
| Moving average (120 m) | 1.14% | 2.66% | 3.85% | 1.03% | 2.56% | 3.66% |
| OLS | 6.46% | 7.89% | 12.04% | 6.43% | 8.16% | 11.57% |
| SIC-3 | 3.60% | 7.22% | 18.62% | 3.85% | 7.51% | 18.00% |
| LASSO | 15.25% | 21.04% | 33.88% | 14.72% | 21.50% | 34.24% |
| Principal components | 1.58% | 2.71% | 6.23% | 1.53% | 2.78% | 6.59% |
| TRF | 2.16% | 3.78% | 5.97% | 2.59% | 3.57% | 6.05% |
| FC MSFE w.avrg. | 2.05% | 3.76% | 6.61% | 1.79% | 3.58% | 6.46% |
| FC VS MSFE w.avrg. | 1.46% | 3.62% | 11.25% | 1.53% | 3.26% | 10.74% |
| Consensus (avrg) | 3.66% | 6.27% | 12.34% | 4.25% | 6.59% | 12.40% |

Notes: This table reports the results of Hansen's (2005) SPA test for BL portfolios (unconstrained) based on multivariate forecast models. Panel A presents the results when using the (negative) portfolio returns as the loss function, whereas Panel B depicts the results for the (negative) risk-adjusted excess returns as the loss function. The evaluation period is from January 1989 to December 2013.

However, the results change when we perform the analysis on a risk-adjusted basis. The OLS and SIC forecasts do not outperform any of the benchmark models, whereas the TRF and principal component forecasts outperform all four benchmark portfolios at significant margins. Important to remember is that the SPA test assesses the null hypothesis that no forecast model outperforms the respective benchmark. We reject this hypothesis for seven out of the eight tests.

Hence, the SPA tests confirm our finding that the forecast models add value to a BL investor and show that our results are robust to data snooping biases. Moreover, the SPA tests indicate that this finding is stronger on a risk-adjusted level and that the TRF and the principal component approaches are among the strongest forecast models.

5.3.5. Performance Measurement with Factor Models

Finally, we analyze whether the forecast-based sector strategy is explained by known risk factors. We regress the returns of the optimized sector portfolios on the five factors proposed by Fama and French (2016), extended by momentum (Carhart 1997) and the betting-against-beta factor (Frazzini and Pederson 2014). The betas of the factor regressions are presented in Table 10.

**Table 10.** Analysis of optimized portfolio returns: Multi-factor regressions.

| Performance Measure | Alpha | Mkt-RF | SMB | HML | RMW | CMA | MOM | BAB |
|---|---|---|---|---|---|---|---|---|
| 1/N | 0.001 | 0.99 ** | −0.142 ** | −0.008 | 0.062 ** | 0.032 | 0.003 | 0.048 ** |
| Cumulative average | 0.002 ** | 0.829 ** | −0.136 ** | 0.032 | 0.029 | −0.008 | −0.021 | 0.078 ** |
| Moving average (60 m) | 0.003 ** | 0.882 ** | −0.202 ** | −0.046 | 0.05 | −0.106 | 0.03 | 0.123 ** |
| OLS | 0.010 | 0.403 | −0.125 | −0.071 | 0.159 | 0.236 | 0.226 ** | −0.309 |
| SIC−3 | 0.007 | 0.581 ** | −0.165 ** | −0.036 | 0.13 | 0.188 | 0.155* | −0.201 |
| LASSO | 0.005 * | 0.542 ** | −0.119 ** | −0.062 | 0.093 | 0.155 | 0.079* | −0.094 |
| Principal components SIC | 0.004 ** | 0.754 ** | −0.173 ** | −0.072 | 0.019 | 0.064 | 0.016 | 0.07* |
| TRF | 0.008 ** | 0.604 ** | −0.143 ** | −0.166 | 0.103 | 0.229 | 0.08 * | −0.064 |
| FC MSFE w.avrg. | 0.003 ** | 0.805 ** | −0.139 ** | 0.007 | 0.034 | 0.016 | −0.01 | 0.066 ** |
| FC VS MSFE w.avrg. | 0.003 ** | 0.777 ** | −0.129 ** | −0.016 | 0.04 | 0.051 | 0.002 | 0.051 ** |
| Consensus (avrg) | 0.001 ** | 0.660 ** | −0.089 ** | 0.031 * | 0.03 * | 0.017 | −0.028 ** | 0.035 ** |

Notes. This table reports multi-factor regressions for the excess returns of the optimized sector portfolios. The portfolios consist of the six U.S. sector indices. Monthly portfolio weights are computed based on the unconstrained Black and Litterman (1992) model. The employed return forecast model for sector indices is given in the first column. The first row depicts the performance of an equally weighted (1/N) portfolio as benchmark strategy. The evaluation period is from January 1989 to December 2013. 'Mkt-RF' is the market risk premium, 'SMB' is the size factor, 'HML' is the value factor, 'RMW' is the quality factor, 'CMA' is the investment factor, 'MOM' is the momentum factor, and 'BAB' is the betting-against-beta factor. 'Alpha' is the monthly percentage excess return above the return, which one would expect based on the multi-factor model. * and ** indicate a statistical significance at the 5% and 1% level, respectively.

Most importantly, we find that all forecast-based strategies except for the simple OLS and SIC models have significant positive multi-factor alphas. Hence, they generate abnormal returns that are not explained by the common risk factors 'market', 'size', 'value' 'quality', 'investment', 'momentum', and 'low risk'. In support of the previously reported traditional performance measures, the target-relevant factor approach (TRF) offers the highest outperformance (0.8% per month).

As expected, all sector allocations have a significant risk exposure to the U.S. stock market. The exposures to the size ('SMB') factor are negative for all strategies because the employed sector indices are market-value weighted and therefore overweigh large-cap stocks relative to small-cap stocks. The value ('HML'), quality ('RMW'), and investment ('CMA') factor exposures are insignificant for all prediction models. The exposure on the momentum factor is slightly positive for most strategies, indicating a minor overweighting of 'winner' stocks. However, this finding is not significant for all forecast models. The betting-against-beta factor (BAB) only plays a significant role in the principal component and the forecast averaging models. These models show a positive exposure, indicating that they tend to overweigh low-beta stocks.

## 6. Robustness Analysis

We conduct a number of additional analyses to test for the robustness of our results. We analyze the portfolio performance in different market environments (1), employ different optimization models (2), and investigate our sector results for a different asset universe based on 17 industries (3).

### 6.1. Performance in Different Market Environments

As a robustness check, we divide the full evaluation period from January 1989 to December 2013 into expansionary and recessionary sub-periods and analyze the sector-level performance in different market environments. Sub-periods are NBER-dated expansions (Exp) and recessions (Rec). Table 11 presents the results. The SIC, the TRF, the

principal component, the forecast combinations, and the consensus forecasts outperform the market and the passive 1/N portfolio in all sub-periods, with the TRF approach yielding the highest performance in four of the seven sub-periods. Overall, the sub-period analysis confirms our results for the full period. Particularly, the outperformance of the forecast models seems to hold in expansionary and recessionary market environments.

**Table 11.** Analysis of sub-periods: Portfolio performance (Sharpe ratios) of BL portfolios with forecasts.

| Period | Jan 89–Aug 90 | Sep 90–Apr 91 | May 91–Apr 01 | May 01–Dec 01 | Jan 02–Jan 08 | Feb 08–Jul 09 | Aug 09–Dec 13 |
|---|---|---|---|---|---|---|---|
| Economic state (NBER) | Exp | Rec | Exp | Rec | Exp | Rec | Exp |
| Length of sub-period (months) | 20 | 8 | 120 | 8 | 73 | 18 | 53 |
| Market | 0.30 | 0.35 | 0.69 | −0.10 | 0.25 | −0.89 | 1.26 |
| 1/N | 0.41 | 0.42 | 0.79 | −0.02 | 0.36 | −0.89 | 1.28 |
| Cumulative average | 0.67 | 0.56 | 1.16 | 0.11 | 0.60 | −0.94 | 1.28 |
| Moving average (60 m) | 0.82 | 0.55 | 1.16 | 0.05 | 0.50 | −1.09 | 1.33 |
| OLS | 0.39 | 0.83 | 1.23 | −0.35 | 0.72 | 0.62 | 1.46 |
| SIC−3 | 0.60 | 0.71 | 1.21 | 0.03 | 0.65 | 0.23 | 1.42 |
| LASSO | 0.47 | 0.76 | 1.23 | −0.05 | 0.62 | 0.38 | 1.44 |
| Principal components | 0.74 | 0.97 | 1.16 | 0.17 | 0.67 | −0.76 | 1.39 |
| TRF | 0.48 | 1.29 | 1.27 | −0.01 | 0.77 | 0.95 | 1.59 |
| FC MSFE w.avrg. | 0.66 | 0.67 | 1.17 | 0.11 | 0.62 | −0.87 | 1.32 |
| FC VS MSFE w.avrg. | 0.65 | 0.79 | 1.20 | 0.07 | 0.61 | −0.70 | 1.34 |
| Consensus (avrg) | 0.58 | 0.86 | 1.24 | −0.01 | 0.68 | 0.29 | 1.44 |

Notes. This table reports the performance (Sharpe ratios) of active asset allocations during sub-periods. The forecast model is given in the first column. Sub-periods are NBER-dated expansions (Exp) and recessions (Rec) during the evaluation period from January 1989 to December 2013. The asset allocations build on the unconstrained Black and Litterman (1992) model. The investment universe includes the six sector indices. The first row shows the Sharpe ratio of the overall U.S. stock market index, and the second row depicts the Sharpe ratio of a passive equally weighted (1/N) sector portfolio as benchmark strategies.

*6.2. Alternative Optimization Models*

As another robustness check, we implement the return predictions in alternative asset allocation models to analyze whether our results depend on the BL model. Table 12 presents the Sharpe ratios for the unconstrained and the short-sale-constrained Markowitz (MV) (1952), Bayes-Stein (BS) (Jorion 1985, 1986), and BL models. The header displays the asset allocation model. In the case of short selling, the BS and MV strategies result in a lower performance for most forecast models. This due to extreme portfolio allocations with large short positions. For the optimization with short-sale constraint, BS and MV models outperform the market portfolio and the passive 1/N strategy for most forecast models. The BL model works well with and without short selling and performs better than the BS and MV models for almost all prediction models. When we disallow short sales, the target-relevant factor forecast provides the highest performance for all asset allocation models. Overall, the results support our major finding that sector-level return forecasts improve portfolio performance over historical averages, passive investments in the market index, and equally weighted portfolios (1/N).

**Table 12.** Portfolio performance (Sharpe ratios) for sector-level forecasts in alternative asset allocation models.

| Asset Allocation Model | BL | BS | MV | BL | BS | MV |
|---|---|---|---|---|---|---|
| | Unconstrained | | | No Short Sales | | |
| Market | 0.46 | 0.46 | 0.46 | 0.46 | 0.46 | 0.46 |
| 1/N | 0.54 | 0.54 | 0.54 | 0.54 | 0.54 | 0.54 |
| Cumulative average | 0.72 ***ˣ | 0.63 | 0.46 | 0.72 ***ˣ | 0.61 | 0.55 |
| Moving average (60 m) | 0.77 *** | 0.56 | 0.35 | 0.76 *** | 0.67 | 0.58 |
| OLS | 0.79 * | −0.05 | 0.00 | 0.89 ***ˣ | 0.75 | 0.76 * |
| SIC−3 | 0.79 ** | 0.00 | 0.09 | 0.87 *** | 0.77 * | 0.85 ** |
| LASSO | 0.93 ***ˣˣ | 0.29 | 0.22 | 0.90 ***ˣˣ | 0.48 | 0.68 |
| Principal components | 0.90 ***ˣˣ | 0.31 | 0.21 | 0.89 ***ˣˣ | 0.73 * | 0.77 * |
| TRF | 1.10 ***ˣˣ+ | 0.47 | 0.31 | 1.03 ***ˣˣ+ | 0.85 * | 0.92 **ˣ |
| FC MSFE w.avrg. | 0.79 ***ˣ | 0.87 ** | 0.81 | 0.79 ***ˣ | 0.77 **ˣ | 0.77* |
| FC VS MSFE w.avrg. | 0.85 ***ˣ | 0.65 | 0.53 | 0.85 ***ˣ | 0.77 * | 0.86 *** |
| Consensus (avrg) | 0.97 ***ˣ | 0.57 | 0.48 | 0.92 ***ˣ | 0.77 * | 0.89 **ˣ |

Notes. This table reports the portfolio performance measures for an investor who allocates monthly between the six analyzed U.S. sectors based on the forecast model given in the first column. The employed asset allocation model is displayed in the header and is either the Black–Litterman (BL) model, the Bayes-Stein (BS) model, or the traditional Markowitz (1952) (MV) model. Each of the three asset allocation models is employed either with or without a short-sale constraint. The investment universe includes the six sector indices. The first row depicts the overall U.S. stock market index and the second row the performance of a passive equally weighted (1/N) sector portfolio as benchmark portfolios. The evaluation period is from January 1989 to December 2013. *,**, (ˣ, ˣˣ), and [⁺,⁺⁺] indicate a statistically significantly higher Sharpe ratio compared to the market (the buy-and-hold 1/N portfolio) [the BL-optimized portfolio based on the cumulative historical average as a return forecast] at the 5% and 1% level, respectively.

*6.3. Alternative Asset Universe*

Additionally, we analyze whether the results for sectors also hold for a more detailed industry universe. Therefore, we repeat our analysis by using the data of the 17 Fama–French industry portfolios and obtain the portfolio returns from the Fama–French website.[20] Earnings, dividends, and trading data are not available for the Fama–French industries, and consequently, the predictor set does not include these variables. We compute monthly return predictions and prediction-based Black–Litterman optimized portfolios. Table 13 presents the forecast accuracy measure $ROS^2$ when using the multivariate forecast models to predict each of the 17 Fama–French industry returns. The evaluation period is from January 1989 to December 2013. The results confirm our base case results, with the TRF approach yielding the highest prediction accuracy ($ROS^2$), followed by the MSFE-weighted combination forecast (FC-MSFE-w.avrg). For 12 (9) of the 17 industries, the forecast errors of the TRF (FC-MSFE-w.avrg) predictions are significantly lower compared to the historical average forecast.

**Table 13.** Alternative asset universe: 17 Fama–French industries—$ROS^2$ of multivariate forecast models.

| Model | Mrkt | Food | Mines | Oil | Clths | Durbl | Chem | Cnsm | Cnstr |
|---|---|---|---|---|---|---|---|---|---|
| OLS | −4.1% | −13% | −0.3% * | −1.8% * | −1.1% ** | −3.3% | −4.4% | −7.5% | −6.5% |
| SIC−3 | −4.9% | −1.9% | 0.8% ** | 2.5% * | −7% | −7.1% | −3.4% | −5.5% | −2.2% |
| LASSO | 0.5% | −3.6% | 2.1% * | 0.5% * | −1.3% | −1.4% | −1.1% | −4.1% | −1.1% |
| Principal components | 0.4% | −0.3% | 0% | 2.2% ** | 0.8% | 1% | 0.9% | −0.1% | 0.7% |
| TRF | 3.7% ** | −8.1% | 0.5% ** | 1.8% ** | 1.4% ** | 0.9% * | 1.4% | −5.5% | −1% * |
| FC-MSFE-w.avrg. | 0.9% ** | −0.1% | 0.6% * | 0.8% ** | 0.6% ** | 0.5% | 0.5% | 0.2% | 0.4% |
| FC-VS-MSFE-w.avrg. | 0.6% | −1.4% | 1.6% * | 1% * | 1.4% ** | 1% * | −0.3% | −1.7% | −1.1% |

| Consensus (avrg) | 1.9% | −1.4% | 2.2% ** | 3% ** | 1.3% * | 0.6% | 0.5% | −1.3% | 0.6% |
|---|---|---|---|---|---|---|---|---|---|
| Model | Steel | FabPr | Machn | Cars | Trans | Utils | Rtail | Finan | Othr |
| OLS | 1.8% * | −2.7% * | −1.1% | −0.3% * | −2.3% * | −9% | −6.8% * | −10% | −1.5% * |
| SIC−3 | 2.5% ** | 0.7% | −0.2% | −2.8% | −2.8% | −0.8% | −6% | −5% | −3.5% |
| LASSO | 2.5% ** | 1% | −0.2% | 1.2% | −0.6% | 0.1% | −4.1% | −1.5% | −0.8% |
| Principal components | 0.2% | 0.9% | 1% | 0% * | 1.1% | 0.7% | 0.2% | 2.1% | 3% ** |
| TRF | 2.3% * | 2.5% ** | 2.1% ** | 3.6% *** | 2.6% ** | −6.3% | −2.7% ** | −0.2% | 2.5% ** |
| FC-MSFE-w.avrg. | 0.6% | 0.7% ** | 0.5% ** | 0.8% *** | 0.7% ** | 0% | 0.4% * | 0.6% | 0.7% ** |
| FC-VS-MSFE-w.avrg. | 1.3% | 1.2% * | 1.2% ** | 1.1% ** | 0.1% | −0.9% | 0.3% | 0.6% | −1% |
| Consensus (avrg) | 2.5% * | 2.5% * | 1.5% * | 2.2% * | 1.8% * | −0.4% | 0.4% | 0.1% | 1.3% * |

Notes. This table provides the evaluation of the forecast error (ROS$^2$) of multivariate forecast models for an alternative universe consisting of 17 Fama–French industries. *, **, and *** indicate statistically significantly better forecasts compared to the historical average at the 10%, 5%, and 1% level, respectively.

In Table 14, we present the performance of the prediction-based monthly BL portfolios for the 17 Fama–French industry portfolios. The results confirm our base case findings that a prediction-based asset allocation strategy adds value for investors and outperforms the market portfolio. In line with the base case results, the principal component regression forecast and the target-relevant factor (TRF) approach yield the highest Sharpe ratios and significantly outperform the market and the 1/N benchmark portfolio.

**Table 14.** Alternative asset universe: 17 Fama–French industries—prediction-based BL portfolios.

| Performance Measure | Return | Vola | Sharpe | Omega | CER | CER-Gain | Turnover | BTC |
|---|---|---|---|---|---|---|---|---|
| Market | 10.39% | 15.14% | 0.46 | 1.41 | 0.05 | 0.00% | 0.00 | |
| 1/N | 12.19% | 14.33% | 0.61 | 1.58 | 0.07 | 2.40% | 0.00 | |
| Cumulative average | 13.52% | 13.02% | 0.78 *× | 1.82 | 0.09 | 4.63% | 2.59 | 1.79% |
| Moving average (60 m) | 13.87% | 13.79% | 0.76 * | 1.83 | 0.09 | 4.47% | 4.13 | 1.08% |
| OLS | 17.39% | 23.63% | 0.59 | 1.99 | 0.03 | −1.23% | 112.33 | −0.01% |
| SIC−3 | 15.23% | 21.31% | 0.56 | 1.85 | 0.04 | −0.79% | 31.35 | −0.03% |
| LASSO | 11.56% | 12.14% | 0.67 | 1.89 | 0.08 | 3.22% | 21.57 | 0.15% |
| Principal components | 16.53% | 11.86% | 1.11 ****× | 2.27 | 0.13 | 8.36% | 5.22 | 1.60% |
| TRF | 20.44% | 16.53% | 1.03 ***× | 2.49 | 0.14 | 8.95% | 23.49 | 0.38% |
| FC MSFE w.avrg. | 14.21% | 12.29% | 0.88 ***× | 1.94 | 0.10 | 5.78% | 2.70 | 2.14% |
| FC VS MSFE w.avrg. | 14.43% | 12.04% | 0.92 ***× | 1.99 | 0.11 | 6.14% | 5.65 | 1.09% |
| Consensus (avrg) | 15.95% | 13.44% | 0.94 * | 2.19 | 0.11 | 6.78% | 13.67 | 0.50% |

Notes. This table reports the portfolio performance measures for an investor who allocates monthly between the 17 Fama–French industries based on the forecast model given in the first column. The asset allocation builds on the unconstrained Black and Litterman (1992) model. The first row shows the performance of the overall U.S. stock market index and the second row the performance of a passive equally weighted (1/N) industry portfolio as benchmark strategies. The evaluation period is from January 1989 to December 2013. 'Return' is the annualized time-series mean of monthly returns, and 'Vola' denotes the associated annualized standard deviation. 'Sharpe' shows the annualized Sharpe ratio and 'Omega' the Omega measures for each portfolio. 'CER' is the certainty-equivalent return. 'CER-Gain' is the gain in CER for an investor who actively invests based on the indicated forecast model compared to an investor who passively invests in the 1/N portfolio. 'BTC' is the break-even transaction cost, which is the level of variable transaction costs per trade for which the active investment strategy based on the forecast model shown in the first column yields the same certainty equivalent return (CER) as the passive market portfolio. *,**, (×, ××), and [⁺,⁺⁺] indicate a statistically significantly higher Sharpe ratio compared to the market (the buy-and-hold 1/N portfolio) [the BL-optimized portfolio based on the cumulative historical average as a return forecast] at the 5% and 1% level, respectively.

Finally, we compare the six-sector and the 17-industry allocation results to gain more insights into whether a more granular industry universe is beneficial. We find that allocations based on 17 industries rather than six sectors do not perform systematically superiorly. When changing from six sectors to 17 industries, the additional diversification advantage is limited, as both universes include a similar number of stocks, and the correlations of industry returns within a sector are high. An essential drawback of a more granular industry universe is that the larger number of parameter estimates increases forecast errors. Moreover, the larger number of industries also increases the portfolio turnover and transaction costs.

## 7. Conclusions

This study provides new evidence on the predictability of stock returns by focusing on sector returns. Based on the idea of information-efficient capital markets and the previous empirical evidence of mutual fund performance, we start with the notion that sector returns are not predictable and that asset allocations based on return prediction models should in general not be able to outperform a passive benchmark portfolio. In the first part, we investigate the predictability of sector stock returns with univariate and multivariate return prediction models (in sample and out of sample). In the second part, we concentrate on optimal asset allocation models based on sector-level return forecasts. We extend the commonly used dataset (Goyal and Welch 2008) by including additional macroeconomic variables such as the initial claims of unemployment, new building permits, the trade-weighted dollar index, and the Chicago Fed National Activity Index. Moreover, we employ popular technical indicators such as momentum and moving average (Neely et al. 2014), as well as the relative strength and the relative strength index (Wilder 1978).

Based on univariate prediction models, we observe that the stock variance, the trade-weighted dollar index, the initial claims of unemployment, and the Chicago Fed National Activity Index have the strongest individual predictive power for sector return forecasts. In addition, we test competing multivariate prediction models such as OLS, selection via the LASSO, principal components, a target-relevant factor approach, and forecast combinations and then propose a variable selection approach that chooses variables based on their ability to forecast returns ahead of the evaluation period. Our empirical results suggest that the LASSO and the target-relevant factor approaches are the most powerful prediction models. The target-relevant factor (TRF) approach generates significantly superior forecasts relative to the historical average for five out of six sectors. Among the predictive variables, the group of macroeconomic variables has the highest predictive power, whereas technical and fundamental variables contain only low predictive power.

In the second part, we evaluate the benefits of the multivariate forecast models for investors when applied in an asset allocation framework. We compare the results to the market portfolio and passive 1/N portfolio, which equally weigh sector indices. Moreover, we include two dynamic portfolio allocations that use the expanding (rolling) historical return average of each sector as return forecasts. While earlier studies build on the Markowitz (1952) optimization framework to analyze the benefits of forecast models, we rely on the Black–Litterman (BL) asset allocation approach. The Markowitz (1952) framework usually performs poorly in portfolios with more than two assets due to estimation error maximization (Michaud 1989), corner solutions (Broadie 1993), and extreme portfolio reallocations (Best and Grauer 1991). In contrast, the Black–Litterman model accounts for uncertainty in return estimates and generates more stable portfolios with a better out-of-sample performance (Bessler and Wolff 2015).

The main findings of our analysis are that implementing sector-level return forecasts in asset allocation decisions results in a superior performance compared to using historical average returns, and it also outperforms the market and a 1/N buy-and-hold portfolio. The outperformance is statistically significant for all tested forecast models. The target-relevant factor (TRF) and the consensus forecasts achieve the highest portfolio

performance. We also provide empirical evidence that our results are robust for expansionary and recessionary sub-periods and alternative portfolio optimization models. When focusing alternatively on the Fama–French industry portfolios, we find that a more granular prediction and allocation over 17 industries rather than over six sectors does not perform systematically superiorly. When switching from six sectors to 17 industries, the relative diversification advantage is limited, as both universes include a similar number of stocks, and the correlations of industry returns within a sector are high. The drawbacks of a more granular industry universe are the higher forecast errors due to the larger number of parameters estimates and the higher transaction costs.

Despite the initial assumption of information-efficient capital markets and the previous empirical evidence that mutual fund managers are hardly able to outperform appropriate benchmarks persistently, we offer the following new insights and significant results. We provide empirical evidence that higher risk-adjusted returns can be achieved, based on our sample period as well as for different sub-periods, for the selected asset classes as well as the employed asset allocation models. Our empirical evidence suggests that first, return prediction models for sectors generate superior return forecasts, and second, including these return predictions in portfolio optimization models such as the Black–Litterman model results in superior asset allocation decisions and consequently in higher risk-adjusted returns for investors.

To determine whether the employed methodologies and approaches will persistently outperform an appropriate benchmark in the future as well as for other asset classes and periods, we need to explore further. Until then, the main hypothesis that performance persistence is difficult to achieve prevails. Asset managers therefore need to be cautious when implementing our approach without further analyses. So far, we limited our study to U.S. sector indices. Including different asset classes such as worldwide stock and bond indices as well as various commodity groups in asset allocation decisions and using return prediction models for all asset classes may result in even higher risk-adjusted returns. We leave analyzing return predictability and portfolio optimization for these asset classes for future research.

**Author Contributions:** Conceptualization, W.B. and D.W.; methodology, W.B. and D.W.; software, W.B. and D.W.; validation, W.B. and D.W.; formal analysis, W.B. and D.W.; investigation, W.B. and D.W; resources, W.B. and D.W.; data curation, W.B. and D.W.; writing—original draft preparation, W.B. and D.W.; writing—review and editing, W.B. and D.W. All authors have read and agreed to the published version of the manuscript.

**Funding:** The authors declare no funding.

**Data Availability Statement:** Restrictions apply to the availability of these data. Data were obtained from Thomson Reuters Datastream and are available from the authors with the permission of Datastream.

**Conflicts of Interest:** The authors declare no conflict of interest.

## Appendix A

**Table A1.** In-sample analysis of univariate predictive power (full sample univariate predictive regression). Amihud and Hurvich (2009): Reduced bias estimation method.

| | Market | | Oil and Gas | | Manufacturing | | Con Gds and Srvcs | | Health Care | | Tech and Tele | | Financials | |
|---|---|---|---|---|---|---|---|---|---|---|---|---|---|---|
| | Coeff | R² | Coeff | R² | Coeff | R² | Coeff | R² | Coeff | R² | Coeff | R² | Coeff | R² |
| | Fundamental Variables and Interest-Rate-Related Variables | | | | | | | | | | | | | |
| DY | −0.059 | 0.32% | 0.005 | 0.01% | −0.081 * | 0.64% | −0.12 *** | 1.85% | −0.018 | 0.03% | −0.019 | 0.03% | −0.106 ** | 1.18% |
| EP | −0.061 | 0.44% | 0.009 | 0.02% | −0.063 * | 0.60% | −0.105 *** | 1.63% | −0.007 | 0.00% | −0.016 | 0.03% | −0.101 *** | 1.39% |
| SVAR | −1.287 *** | 1.93% | −0.761 *** | 1.35% | −0.938 ** | 0.92% | −0.999 ** | 0.91% | −0.724 | 0.41% | −1.484 *** | 2.71% | −1.025 *** | 2.73% |
| LTR | 0.105 | 0.53% | 0.008 | 0.00% | 0.112 * | 0.57% | 0.141 * | 0.74% | 0.125 * | 0.73% | 0.087 | 0.22% | 0.177 ** | 0.93% |
| TMS | 0.208 | 0.47% | 0.134 | 0.12% | 0.183 | 0.34% | 0.284 * | 0.66% | 0.065 | 0.05% | 0.3 * | 0.59% | 0.161 | 0.18% |

| | | | | | | | | | | | | | | |
|---|---|---|---|---|---|---|---|---|---|---|---|---|---|---|
| DFR | 0.347 ** | 1.27% | 0.437 ** | 1.35% | 0.371 ** | 1.37% | 0.403 ** | 1.31% | 0.348 ** | 1.25% | 0.177 | 0.20% | 0.332 * | 0.71% |
| Macroeconomic Variables | | | | | | | | | | | | | | |
| INFL | −0.16 | 0.01% | −0.88 | 0.28% | 0.014 | 0.00% | −0.256 | 0.03% | −0.277 | 0.04% | −0.706 | 0.16% | 1.162 | 0.47% |
| CLAIMS | −0.087 ** | 0.84% | −0.038 | 0.11% | −0.107 ** | 1.19% | −0.126 ** | 1.33% | 0.022 | 0.05% | −0.122 ** | 0.99% | −0.072 | 0.36% |
| INDP | 0.556 ** | 0.82% | 1.193 *** | 2.53% | 0.686 ** | 1.18% | 0.006 | 0.00% | 0.347 | 0.32% | 0.496 | 0.40% | 0.486 | 0.39% |
| CFNAI | 0.007 *** | 2.11% | 0.006 ** | 1.28% | 0.007 *** | 2.15% | 0.006 ** | 1.21% | 0.002 | 0.29% | 0.009 *** | 2.20% | 0.009 *** | 2.38% |
| PERMIT | 0.051 | 0.46% | 0.042 | 0.22% | 0.061 * | 0.63% | 0.072 * | 0.71% | 0.015 | 0.04% | 0.043 | 0.21% | 0.079 * | 0.69% |
| FEX | −0.365 *** | 1.89% | −0.524 *** | 2.60% | −0.399 *** | 2.13% | −0.157 | 0.27% | −0.315 *** | 1.38% | −0.419 *** | 1.50% | −0.267 * | 0.62% |
| OIL | −0.032 * | 0.60% | −0.005 | 0.01% | −0.014 | 0.12% | −0.051 ** | 1.17% | −0.027 | 0.41% | −0.056 ** | 1.14% | −0.024 | 0.20% |
| Technical Indicators | | | | | | | | | | | | | | |
| MA | 0.009 * | 0.66% | 0.001 | 0.00% | 0.011 ** | 0.88% | 0.005 | 0.13% | 0.002 | 0.03% | 0.01 | 0.48% | 0.006 | 0.19% |
| MOM | 0.01 * | 0.72% | 0.003 | 0.03% | 0.006 | 0.20% | 0.005 | 0.18% | 0.008 * | 0.54% | 0.01 | 0.40% | 0.007 | 0.23% |
| VOL | 0.004 | 0.21% | 0.004 | 0.14% | 0.006 | 0.33% | 0.012 ** | 1.25% | 0.001 | 0.02% | 0.001 | 0.01% | 0.014 ** | 1.31% |
| RSI | 0.007 | 0.04% | −0.015 | 0.13% | 0.008 | 0.06% | 0.033 ** | 0.78% | 0.002 | 0.00% | −0.002 | 0.00% | 0.033 * | 0.63% |
| RS | - | - | −0.315 ** | 1.01% | −0.415 | 0.39% | −0.397 | 0.62% | −0.16 | 0.20% | 0.648 *** | 1.77% | 0.154 | 0.07% |
| MARKET | 0.067 | 0.42% | 0.043 | 0.11% | 0.065 | 0.37% | 0.125 ** | 1.16% | 0.013 | 0.01% | 0.031 | 0.05% | 0.129 ** | 0.98% |

Notes*:* This table provides the regression coefficients and R-squared statistics of in-sample univariate predictive regressions over the full period from January 1974 to December 2013 for the six sectors and the market. We analyze 19 predictive variables, including fundamental variables and interest-rate-related variables, macroeconomic variables, and technical indicators. *, **, and *** indicate values significantly different from 0 at the 10%, 5%, and 1% level, respectively.

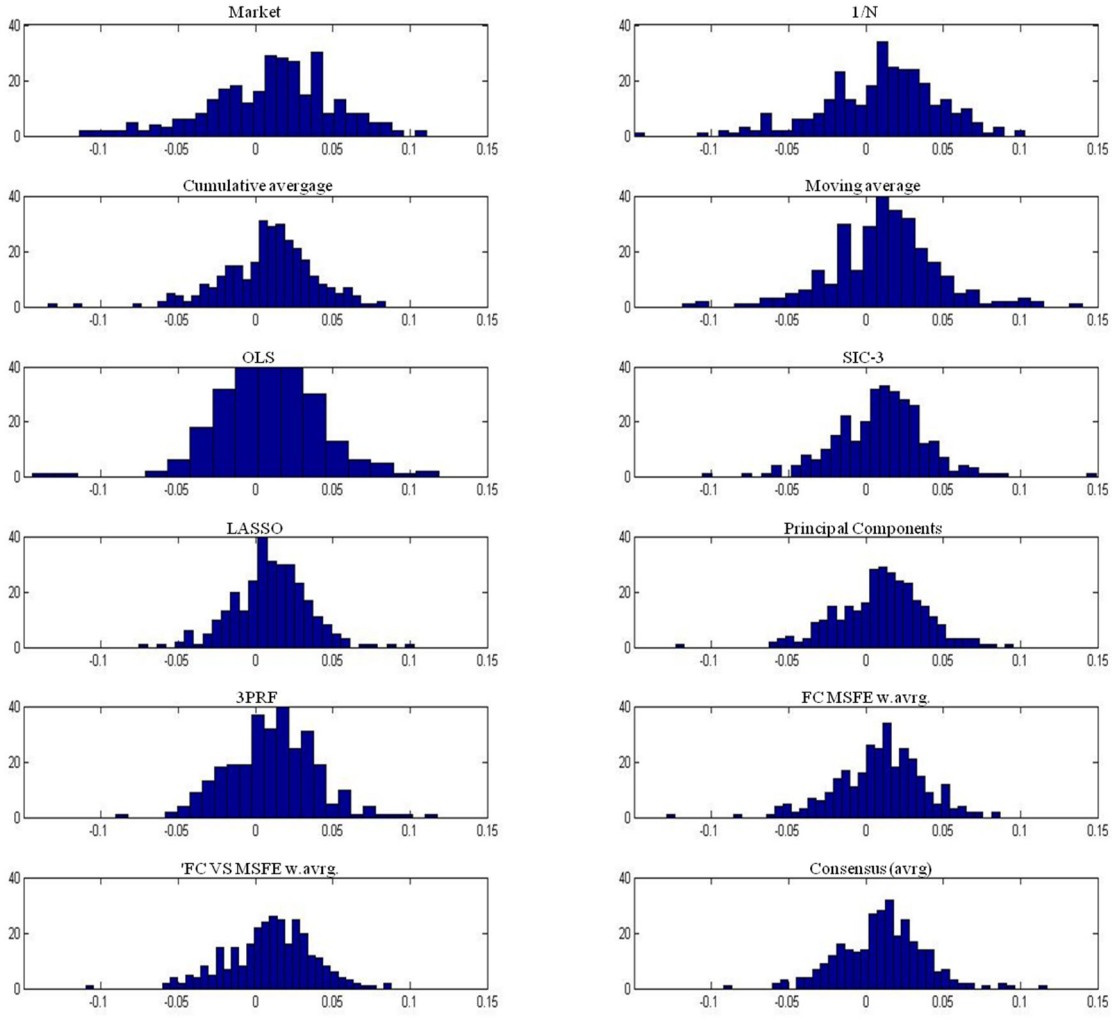

**Figure A1.** Distribution of returns. Notes: This figure shows the distribution of BL-optimized portfolio returns during the full period from January 1989 to December 2013 using the indicated return forecast models.

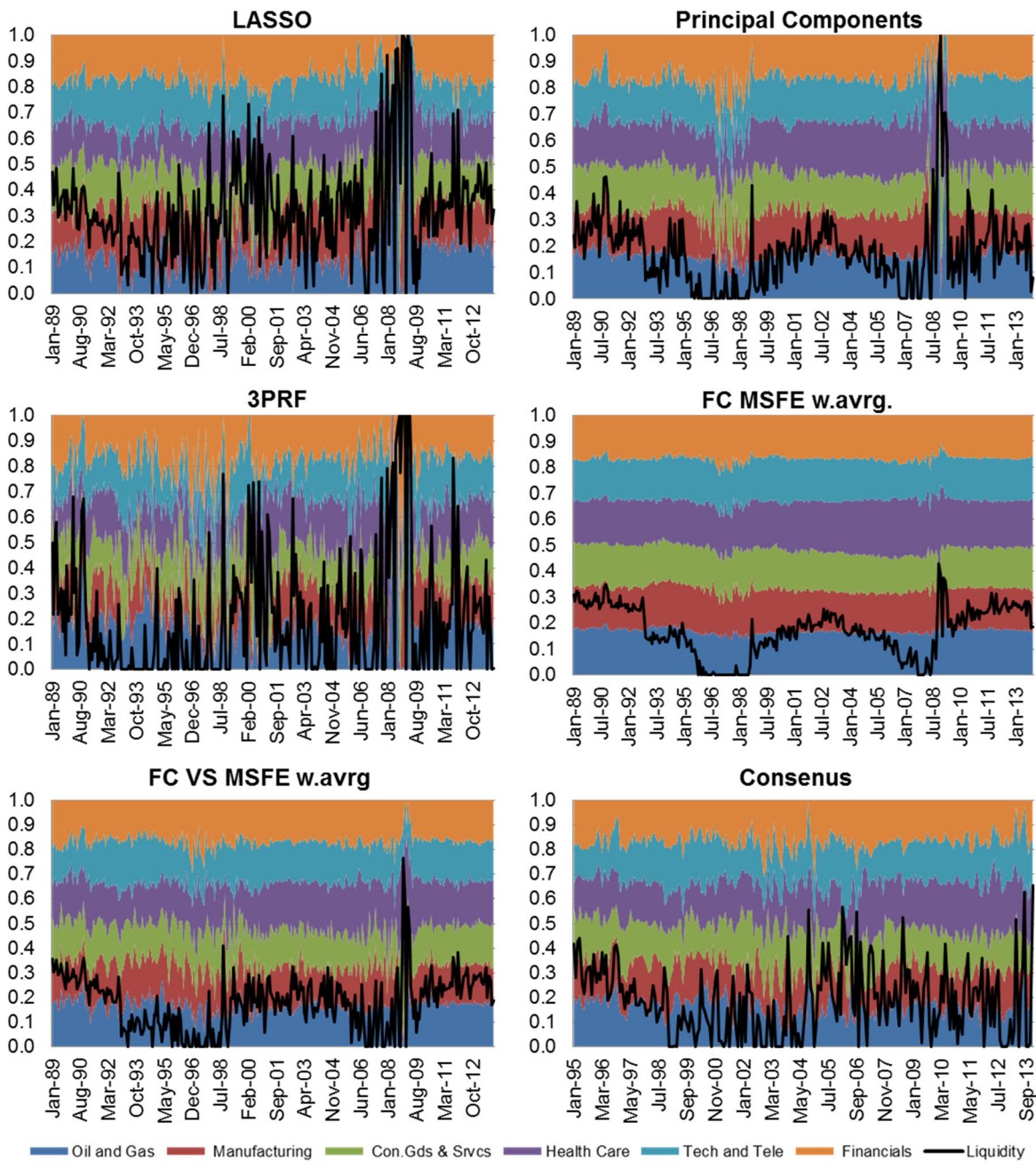

**Figure A2.** Allocations over time. Notes*: This figure shows the portfolio allocation of BL-optimized portfolios over the full period from January 1989 to December 2013 using the indicated return forecast models.

## Notes

1. The BL model also allows us to stay neutral for some assets and does not require forecasts for these assets. In our analysis, we compute forecasts for all assets.
2. Rapach and Zhou (2013) provide a literature review on forecasting stock returns.
3. The dividend yield (Dow 1920; Fama and French 1988; Ang and Bekaert 2007), the book-to-market ratio (Kothari and Shanken 1997; Pontiff and Schall 1998), the term spread (Campbell 1987), the default spread (Keim and Stambaugh 1986; Fama and French 1989), the price-to-earnings ratio (Fama and French 1989; Zorn and Lawrenz 2017), the inflation rate (Nelson 1976; Fama 1981), the stock variance (Guo 2006), the world's capital-to-output ratio (Cooper and Priestley 2013), the difference between the dividend yield and the 10-year treasury bond yield (Maio 2013), and inventory productivity (Alan et al. 2014).
4. For instance, it is possible that the encountered relation between a predictive variable and future returns is not stable over time and therefore cannot be utilized to improve performance. Moreover, transaction costs might hinder investors from exploiting predictability.

5.  Combinations are either simple averages of forecasts or weighted averages based on the forecast performance during a holdout period. Both approaches have significant out-of-sample predictive power for forecasting the S&P500 during the 1951 to 2011 period (Rapach et al. 2010).
6.  For instance, in the sense that coefficients in predictive regressions are set to zero if they do not match the theoretically expected sign, thereby reducing estimation error and improving out-of-sample forecast performance.
7.  The basic idea of using principal components for return prediction is to extract a smaller set of uncorrelated factors of a usually large set of correlated predictors, thereby filtering out noise.
8.  The Thomson Reuters Datastream computes ten sector indices. We aggregate related sectors in order to reduce complexity. More precisely, our 'Manufacturing' index comprises the Datastream indices 'Basic Materials', 'Industrials', and 'Utilities'. Our 'Consumer Goods and Services' index comprises the Datastream indices 'Consumer Goods' and 'Consumer Services.' Our 'Technology and Telecommunication' index comprises the Datastream indices 'Technology' and 'Telecommunication'. We aggregate indices computing market-value-weighted returns and market-value-weighted fundamental data.
9.  The extended Goyal and Welch (2008) dataset is available at https://www.ivo-welch.info/professional/goyal-welch/ (accessed on 10 November 2023). We cannot use the full Goyal and Welch (2008) dataset, as it contains mostly market-wide factors rather than industry-specific variables. The excluded fundamental variables due to data restrictions include the corporate equity activity, the book-to-market ratio, and the dividend payout ratio. Moreover, we do not include bond yields (long-term yield and default yield), as we expect the information of these variables to be already captured in the employed bond returns (long-term return and default return spread).
10. The macroeconomic data are from the St. Louis Fed's FRED database. http://research.stlouisfed.org/fred2/ (accessed on 10 November 2023).
11. The CFNAI is constructed to have an average value of zero and a standard deviation of one.
12. One monthly observation is lost due to the distinct time index on both sides of the predictive regression in Equation (4).
13. More precisely, we test the null hypothesis that the the historic average MSFE is less than or equal to the MSFE of the prediction model against the one-sided alternative hypothesis that the historic average MSFE is greater than the MSFE of the prediction model. The Clark and West (2007) MSFE-adjusted statistic allows for the testing of statistically significant differences in forecasts from nested models. It is computed as the sample average of $f_{t+1}$, and its statistical significance can be computed using a one-sided upper-tail t-test:
$$f_t = (r_t - \bar{r}_t)^2 - [(r_t - \hat{r}_t)^2 - (\bar{r}_t - \hat{r}_t)^2]$$
14. This stylized fact is termed the 'forecast combining puzzle' because in theory, it should be possible to improve upon simple combination forecasts.
15. MSFE-based weighted combination forecasts are equivalent to optimal combination forecasts when correlations between different forecasts are neglected.
16. For alternative risk aversion coefficients of 2 and 10, we obtain similar results.
17. This is in line with Bessler and Wolff (2015) and Bessler et al. (2017). Earlier studies use similar values ranging from 0.025 to 0.300 (Black and Litterman 1992; Idzorek 2005). Bessler et al. (2017) provide evidence that the performance of the Black–Litterman model is quite robust for different parameter choices.
18. This test is applicable under very general conditions—stationary and ergodic returns. Most importantly for our analysis, the test permits the auto-correlation and non-normal distribution of returns and allows for correlation between the portfolio returns.
19. We use the data from Kenneth French's website, which is available at http://mba.tuck.dartmouth.edu/pages/faculty/ken.french/data_library.html#Research (accessed on 10 November 2023).
20. The data is available at http://mba.tuck.dartmouth.edu/pages/faculty/ken.french/data_library.html#Research (accessed on 10 November 2023).

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
