# Peer review of "Portfolio Optimization with Sector Return Prediction Models"

_jrfm, doi:10.3390/jrfm17060254_

Round 1

Reviewer 1 Report

Comments and Suggestions for Authors

This paper uses several multilinear regression models to model monthly sector returns against a range of predictors, both financial and macroeconomic variables. Then, the authors use these sector-by-sector forecasts, feed them into the Black-Litterman model, and yield portfolio weights. They compare these portfolios with some benchmark portfolios such as the 1/N weighted portfolio. The paper is long and thorough (perhaps even too long), with detailed regression experiments, validation of their results, and robustness checks. For the most part, the paper is very good, and should certainly be acceptable for publication following some revisions.

1) The quality of writing is generally good and clear, but there are minor typos throughout. Verb tense and singular/plural are inconsistent. Many commas are missing that would make long sentences clearer. And there are some misc typos. A list is included at the end of this review. This is a non-exhaustive list, please read the manuscript closely.

2) I had numerous issues with the tables. Tables are inconsistently named eg Table II in the text vs Table 2 at the table and inconsistently capitalized eg Table II but table IV. You reference “Table VII provides” when I think you mean Table 7a. Having said that, I don’t like Table 7a and 7b, just make them all whole numbers please. For example, Table VIb should be its own Table 7 and placed after the discussion of the SPA-test - it’s not that closely related to Table VIa. I think it would only make sense to have Table 1a and 1b if the two were to record the exact analogous quantities, just for different settings.

More importantly, Table 4 is very hard to read due to percentages on different lines, and minus signs on different lines. Consider marking numbers in the table with color. Eg blue if R^2_os > 0.5 and red if negative.

The methods in Table 6 need a glossary, perhaps a smaller table with a description. For example, I wasn’t sure at first that FC VS-MSFE referred to the variable selection method described in the first paragraph of page 20, as the acronym was not used before the table.

“Vola” in Table 7 is informal. I know you’re short on width, try shrinking the width of other columns. 

I don’t quite understand the BTC in Table 7a. “Investors could pay a management fee as high as 846 (559) basis points per year and 741 still benefit from active investing based on the TRF” However the BTC for the TRF is recorded as 1.19%. 

“Panel A shows the results compared to the historical average 786 forecast in the out-of-sample period from January 1989 to December 2013 for a loss function based 787 on mean squared forecast errors.” For completeness, this caption should also describe Panel B.

“[Table VIIb about here]” What is this?

“For the Oil & Gas (Consumer Goods 545 & Services) sector, seven (six) out of eight prediction models significantly outperform the 546 historical average in forecasting future returns.” This doesn’t seem to be correct for consumer goods and services, where I count just five models where R^2_os is positive. Again, it’s very hard to read because the minus sign is printed above the rest of the number. Clearly define what you mean here, do you mean R^2_os is positive, or the t-statistic is significant (at what threshold) or both?

3) Your paper lacks recent referencing. There is only one cited paper more recent than 2018 (from 2020). Your paper also lacks referencing specifically on the topic of portfolio optimization. There has been a great deal of recent work using new approaches (such as machine learning algorithms) for forecasting and portfolio optimization. Please cite the following at a minimum and look elsewhere for recent work: “LSTM-based Deep Learning Model for Stock Prediction and Predictive Optimization Model” (https://doi.org/10.1016/j.ejdp.2021.100001), “Semi-Metric Portfolio Optimization: A New Algorithm Reducing Simultaneous Asset Shocks” (https://doi.org/10.3390/econometrics11010008) “Stock Market Forecasting Based on Spatiotemporal Deep Learning” (https://doi.org/10.3390/e25091326)

4) Clarifying questions and presentation issues:

In the body, you refer to bivariate regression. In the appendix, you refer to univariate regression. Please correct me if I am wrong, but this is just the same thing, the simple OLS linear regression of one response against one predictor. If these are the same thing, standardize terminology. If they are different, please explain the difference.

The Goyal and Welch dataset URL doesn’t work - it says it has been moved. Given this resource is from 2008, how was it used for your data up to 2013? Was it updated? Please provide a new URL.

“autocorrelation is not a concern for most variables and no autocorre- lation coefficient exceeds 0.95.” 0.95 is very high, this doesn’t justify for me that autocorr is not a concern? 

“Consequently, the SPA-test detects whether there is at least one superior forecast model.” “Important to remember is that the SPA test assesses the null hypoth-810 esis that no forecast model outperforms the respective benchmark.” I’m not familiar with this test, but it seems you use it on one model at a time? So I don’t quite understand this.

In Section 4, what is r_t exactly? Monthly log returns for the entire market and the six sectors? This must be more clearly presented.

“We use the monthly 638 sector-level return forecasts resulting from multivariate regression models to compute 639 monthly BL-optimized portfolios.” Details on how this works? Going back to Section 2, “in which Pi is the vector of implied asset returns, Sigma is the covariance matrix, and Q is 116 the vector of the return estimates” what is the difference between Pi and Q? Just to clarify, Pi is computed just from the reference portfolio, so you use the forecasts from your regression models into Q?

5) Statistical issues

“Simply picking the best variables for each sector based on the forecast-ability for the overall sample clearly incorporates a look-ahead bias and therefore is not adequate for

our out-of-sample ex ante approach.” This is a minor point and probably doesn’t matter, but in theory, couldn’t you do a validation set approach to avoid look-ahead bias? So a different explanation might be due here.

“Variables are se-526 lected on an ex ante basis. That is, for the decision whether a specific variable is included 527 in forecasting the return for the subsequent period (t+1), we rely on the bivariate predic-528 tive regression including returns until the current period [0, t]. A variable is only included 529 in the combination forecast if its regression coefficient estimate is significant at the 10%-530 level, using robust (Newey-West) standard errors.” 

This strikes me as a misuse of p-values. p-values are frequently overused, and should generally only be used to retain or reject a clearly stated hypothesis, not for feature selection. In addition, there is no adjustment for multiple hypotheses here.

“Pre-selecting relevant variables before com-573 puting a forecast combination does not seem to add value relative to the forecast combi-574 nation model that simply includes all variables. For all sectors, the variable selection 575 model (FC-VS-MSFE-w.avrg) provides lower adjusted MSFE-statistics than the forecast 576 combination model with all variables (FC-MSFE-w.avrg).” As I am skeptical of this method’s use of p-values and it doesn’t seem to help either, I would consider dropping it entirely. Consider applying an alternative approach to variable selection here. As FC-MSFE is closely related to FC-VS-MSFE, dropping the latter may even improve the consensus forecast, perhaps. However, this is just an idea.

More broadly, throughout the paper, you are consistently testing multiple hypotheses at once, and should consider adjusting your p-values.

6) Misc

Footnotes: personally, I dislike footnotes, as do a growing number of people these days. It’s quite common that footnotes are not allowed in mathematical, scientific or statistical papers. The MDPI style guide states “Most scientific journals from MDPI do not allow footnotes, however those in humanities and some areas of business and economics do.” As this paper is quite mathematical in content (albeit, yes, a financial/economic paper), I recommend eliminating all the footnotes, and absorbing them into the text.

For example, “Rapach and Zhou (2013) provide a literature review on forecasting stock returns” could go in manuscript for example, either as a sentence or a single citation. I won’t force you to remove the footnotes as perhaps it’s your personal taste, but I strongly suggest you consider reducing your usage of them, and incorporating some into the text. Some of the footnotes contain content that is quite important, eg footnote 3, and could be communicated better in a table.

Section 4.1 and 4.2: I think these sections are too long, and quite a bit of space is devoted to a summary of which coefficients are positive/negative in line with well-known economic principles. I don’t think this section is very original, not nearly as interesting as the rest of your paper. In addition, this section also isn’t used later on. I would consider dropping these sections entirely or drastically shortening the discussion. Alternatively, you could move it into the appendix with the univariate regression experiments A1 that are closely related.

Minor English issues and typos:

Decapitalize: Index Funds

“As robustness check,” As a robustness check

“Section 6 concludes” you conclude in Section 7.

“find that for historical average return forecasts no optimization model” ->find that for historical average return forecasts, no optimization model

“To compute technical indicators one year of data is required so that our evaluation period for the in-sample analysis ranges from January 1974 to December 2013 (480 monthly observations).” Add commas for readability

Standardize present vs past tense throughout the manuscript.

“Kelly and Pruitt (2013, 2015) extend 494

PLS to what they call a three-pass regression filter for estimating target-relevant factors 495

(TRF). In a simulation study and two empirical applications Kelly and Pruitt (2015) pro- 496

vides evidence” Inconsistent plural vs singular.

“Null-hypotheses” no hyphen. 

“Fundamental and interest rate related variables: Interestingly, the valuation ratios 299 dividend-yield and the earnings-price ratio only have significant predictive power for the 300 Financial and the Consumer Goods & Services sectors.” This sentence seems to come out of nowhere. It’s also informal to start a sentence like that.

“meas-ure of return predictability” typo

“statistical significant levels” statistically

“The market portfolio, a passive 1/N buy-and-hold 708 portfolio, which equally weighs all six sector-indices and two BL-optimized portfolios” add comma for readability

“Compared to the market and the 1/N portfolio all allocations based on return forecast models enhance the portfolio return.” add comma

“propsed byFama and French (2015) extended by Momentum” typo and add commas

“returns which is not explained” are not

“The exposure to the size (‘SMB’) factor are negative” exposure is negative or exposures are negative

“We compare the results to the 982 market portfolio, a passive 1/N portfolio, which equally weighs sector-indices and two 983 dynamic portfolio allocations, which use the expanding (rolling) historical return average 984 of each sector as return forecasts.” Two consecutive which clauses, unclear.

“When switching from six sectors to 17 industries the relative”->When switching from six sectors to 17 industries, the relative

Comments on the Quality of English Language

Some typos and minor issues, as detailed in the review. Paper is mostly very well written.

Author Response

  • Quality of Writing: We thank the referee for reading the paper very carefully and indicating some shortcomings. We carefully edited the document and improved the writing of the paper and cross-checked for typos and commas.

  • Tables: Thank you very much for looking very carefully through all tables. We consistently capitalized all tables and made all tables whole numbers. We also agree that table 4 was very hard to read. Therefore, we excluded the R^2 measures from the table.

We agree that the models require a glossary and added it to table 6 and table 7 to explain all multivariate regression models.

We agree that “Vola” in table 7 (now table 8) is informal and corrected it to volatility.

Break-even-transaction costs: The statement that you cited “investors could pay a management fee as high as 846 (559) basis points per year and still benefit from active investing based on the TRF (LASSO) forecast model” refers to the gain in certainty equivalent return (CER-Gain) before transaction costs. We clarified the statement in the text and divided the chapter into two sections “Gain in certainty equivalent return (CER-Gain)” and “Turnover and break-even transaction costs (BTC)”.

Caption in table 9: We improved the description of table 9 and adjusted the caption.

Significance levels in table 6: The statement “For the Oil & Gas (Consumer Goods & Services) sector, seven (six) out of eight prediction models significantly outperform the historical average in forecasting future returns, as indicated by the significant t-statistic at the 10%-level” refers to the t-statistic rather that to the out-of-sample R^2. However, we clarified that in the text. For some models the  statistic is negative while at the same time the MSFE-adjusted t-statistic is positive. This is a well-known phenomenon and stems from the fact that the MSFE-adjusted statistic accounts for the higher volatility in predictive regression forecasts compared to the historical average (Clark and West, 2007; Neely et al., 2014).

  • Literature: Thank you very much for suggesting additional and newer literature. We included the suggested papers and added a number of other relevant papers related to equity market predictions and portfolio optimization that have been published during the recent years.

  • Clarifying questions and presentation issues

Bivariate vs. univariate predictive regression. You are completely right. Both refer to a model with one predictor and one response. We corrected the paper and standardized the terminology to univariate predictive regressions.

Goyal/Welch dataset:  We updated the link to the Goyal/Welch dataset.

Autocorrelation: We agree that the statement “autocorrelation is not a concern for most variables” lacks empirical evidence and deleted it from the paper.

SPA-test: The SPA-test allows a comprehensive testing of several forecast models and ensures that the results are robust to biases from data snooping. The SPA-test builds on White’s (2000) “reality check” (RC) but is more powerful and less sensitive to the inclusion of poor and irrelevant alternatives. Consequently, the SPA-test detects whether there is at least one superior forecast model. However, it is not able to identify all superior strategies. Therefore, the SPA-test tests the null hypothesis that the benchmark is not inferior to any alternative forecast model. The results reject the null hypothesis, indicating that at least one forecast model outperforms the historical average even after controlling for data mining

Definition of rt: We added a description of rt in section 4.

Black-Litterman optimization: Your understanding is correct. Pi is the implied return from the reference portfolio (1/N) and Q is the forecast output using our prediction models.

  • Statistical Issues

The statement: “Simply picking the best variables for each sector based on the forecast-ability for the entire sample incorporates a look-ahead bias and therefore is not adequate for our out-of-sample ex ante approach.” We agree that your proposed validation set approach would work. However, picking the best indicators based on the full dataset would include a look ahead bias. Therefore, we only select indicators based on the training set.

P-values for data selection: We completely agree that p-values are not adequate for data selection. Therefore, we also include the Schwarz information criterion for selecting indicators. However, in practice and academia p-values are often misused for identification of relevant data point. Therefore, we include this practice in our analysis and add a comment in the text.

  • Misc

Footnotes: We prefer to keep footnotes. Including the content of the footnoted into the text would disturb fluent reading.

Minor English issues and typos: We corrected all typos and commas.

Reviewer 2 Report

Comments and Suggestions for Authors

This paper offers a valuable analysis of stock return predictability. The literature review presents a critical view of portfolio optimization methods and sector return forecasts, the analysis is accurately illustrated and the results are sound. I am in favour of acceptance. My only concern is on the range of data which terminate in Dec. 2013, that is, more recent data are not considered. The authors should add a justification of their choice of the data period. 

Minor remarks

p.2, line 72: write ‘We separate our empirical analysis into…..’

p.3, line 126: the symbol ϖ is not the same as in (8)

p. 10-11: the correlation matrix is split between two pages. I understand that its size is huge, but would it be possible to concentrate it in a single page – maybe using smaller characters ?

Reference item 77: add a specification or a link to ‘Working Paper’

Author Response

Data limitation: Unfortunately, due to data license restrictions we are not able update our dataset.

p.2, line 72: We adjusted the sentence as suggested.

P3, line 126: We changed the symbol  to be consistent in the formula and in the description.

Table 3: We adjusted the correlation matrix to fit on one page.

Reference item 77: We added a link to the working paper.

Round 2

Reviewer 1 Report

Comments and Suggestions for Authors

Many thanks to the authors for their revisions. I admittedly find the marked manuscript quite hard to read, but it looks like the authors have covered everything, so I don't need to see the manuscript again. I noticed a few typos, such as

"we let the Schwarz information criterion (SIC) decide on the best model in that we allow up to three predictors of any combination in the model. At each point in time, we calculate the Schwarz's Bayesian Information Criterion"

There is inconsistent capitalization and terminology here and an awkward the before Schwarz's. I'm sure MDPI copyeditors will find more typos, but just try to make everything consistent.

Comments on the Quality of English Language

See previous